# PNKP is required for maintaining the integrity of progenitor cell populations in adult mice

Wisoo Shin[1,*], Whitney Alpaugh[1,*], Laura J Hallihan[2,3,*], Sarthak Sinha[1], Emilie Crowther[1], Gary R Martin[2,3], Teresa Scheidl-Yee[2], Xiaoyan Yang[4], Grace Yoon[1], Taylor Goldsmith[1], Nelson D Berger[3], Luiz GN de Almeida[2,3], Antoine Dufour[2,3,5], Ina Dobrinski[1], Michael Weinfeld[4], Frank R Jirik[2,3,6], Jeff Biernaskie[1,5,7,8]

DNA repair proteins are critical to the maintenance of genomic integrity. Specific types of genotoxic factors, including reactive oxygen species generated during normal cellular metabolism or as a result of exposure to exogenous oxidative agents, frequently leads to "ragged" single-strand DNA breaks. The latter exhibits abnormal free DNA ends containing either a 5′-hydroxyl or 3′-phosphate requiring correction by the dual function enzyme, polynucleotide kinase phosphatase (PNKP), before DNA polymerase and ligation reactions can occur to seal the break. *Pnkp* gene deletion during early murine development leads to lethality; in contrast, the role of PNKP in adult mice is unknown. To investigate the latter, we used an inducible conditional mutagenesis approach to cause global disruption of the *Pnkp* gene in adult mice. This resulted in a premature aging-like phenotype, characterized by impaired growth of hair follicles, seminiferous tubules, and neural progenitor cell populations. These results point to an important role for PNKP in maintaining the normal growth and survival of these murine progenitor populations.

## Introduction

To maintain genomic integrity, DNA damage response (DDR) mechanisms actively recognize and correct up to 70,000 lesions per cell per day (Tubbs & Nussenzweig, 2017). Preserving efficient and competent DDR mechanisms are vital for normal development and tissue maintenance throughout life (Hoeijmakers, 2009). Polynucleotide kinase phosphatase (PNKP) is a bifunctional DNA-processing enzyme active during both single- and double-strand break repair (SSBR and DSBR) (Jilani et al, 1999; Karimi-Busheri et al, 1999; Weinfeld et al, 2011). PNKP is recruited to sites of DNA damage by XRCC1 during base-excision repair and by XRCC4 during nonhomologous end joining (NHEJ) via interactions through its Forkhead-associated domain (FHA) (Koch et al, 2004; Loizou et al, 2004; Bernstein et al, 2005; Aceytuno et al, 2017). Once recruited, the 3′-phosphatase and 5′-kinase domains of PNKP remove either the 3′-phosphate, or phosphorylate the 5′-hydroxyl group, respectively, to prepare the free ends for ligation by either DNA ligase 3 or 4 (Whitehouse et al, 2001; Weinfeld et al, 2011). PNKP is required for SSBR, as well as DSBR via NHEJ (Whitehouse et al, 2001; Shimada et al, 2015; Jiang et al, 2017). In addition, PNKP is also present within mitochondria where it is critical for the repair of oxidative mitochondrial DNA (mtDNA) lesions within these organelles (Mandal et al, 2012; Tahbaz et al, 2012).

In humans, autosomal recessive or compound heterozygous *PNKP* mutations are responsible for a variety of neurodevelopmental disorders, including microcephaly with early onset seizures and developmental delay (MCSZ) (Shen et al, 2010; Nakashima et al, 2014; Kalasova et al, 2019), a neurodegenerative disease known as recessive ataxia with oculomotor apraxia type 4 (AOA-4) (Bras et al, 2015), and variants of Charcot–Marie–Tooth disease (Pedroso et al, 2015; Leal et al, 2018). Alterations in *PNKP* expression levels in humans have also been linked to the pathogenesis of spinocerebellar ataxia type 3 (SCA3) in combination with mutant *Ataxin-3* (*ATXN3*) (Chatterjee et al, 2015; Gao et al, 2015). Mutations underlying the neurological disorders are primarily found within the kinase domain of PNKP, although a few have been identified in either the phosphatase or the FHA domains (Shen et al, 2010; Poulton et al, 2013; Nakashima et al, 2014; Kalasova et al, 2019). Despite being clinically distinct diseases, AOA4 and MCSZ mutations are primarily point mutations spread throughout these three domains, with no obvious genotype-to-phenotype correlation (Dumitrache & McKinnon, 2017). Mutations in human *PNKP* can reduce the stability of the encoded protein, leading to reductions in both the 3′-phosphatase and the 5′-kinase activities and protein levels by as much as 10-fold (Shen et al, 2010; Reynolds et al, 2012).

[1]Department of Comparative Biology and Experimental Medicine, University of Calgary, Calgary, Canada   [2]McCaig Institute for Bone and Joint Health, Calgary, Canada   [3]Department of Biochemistry and Molecular Biology, University of Calgary, Calgary, Canada   [4]Department of Oncology, University of Alberta, and Cross Cancer Institute, Edmonton, Canada   [5]Department of Physiology and Pharmacology, University of Calgary, Calgary, Canada   [6]Alberta Children's Hospital Research Institute, Calgary, Canada   [7]Department of Surgery, University of Calgary, Calgary, Canada   [8]Hotchkiss Brain Institute, Calgary, Canada

Correspondence: jirik@ucalgary.ca; jeff.biernaskie@ucalgary.ca
*Wisoo Shin, Whitney Alpaugh, and Laura J Hallihan contributed equally to this work

Studies in mice have demonstrated that global knockout of *Pnkp* is embryonically lethal, such that conditional inactivation of floxed *Pnkp* with the Sox2-Cre driver led to death by embryonic day 9 and restricting deletion of floxed *Pnkp* to neural lineages with the Nestin-Cre driver, led to postnatal death by day 5 (Shimada et al, 2015). These studies highlighted the critical role for *Pnkp* during development. Furthermore, Shimada et al (2015) carried out elegant comparative studies of both the Nestin-Cre:*Pnkp*$^{fl/fl}$ mice (between P2 and P5) and the tamoxifen-inducible glial fibrillary acidic protein (GFAP)-CreER$^{T2}$ driver in *Pnkp*$^{fl/fl}$ mice. These experiments demonstrated that the loss of *Pnkp* compromised neuro-progenitor function, as shown by greatly diminished neurogenesis and oligodendrogenesis (Shimada et al, 2015).

*Pnkp* mutations in humans have been associated with a variety of neurological disease phenotypes, and transgenic mouse model studies have highlighted the neurodevelopmental or neurodegenerative defects that result from *Pnkp* loss (Shen et al, 2010; Nakashima et al, 2014; Shimada et al, 2015; Dumitrache & McKinnon, 2017; Bermúdez-Guzmán & Leal, 2019; Gatti et al, 2019; Kalasova et al, 2019). The finding that *Pnkp* mutations interfere with neuro-development and CNS function is not unprecedented because various mutations in DDR genes, and especially genes involved in base-excision repair and ssDNA break repair, have been linked to a variety of neurological disorders (Dumitrache & McKinnon, 2017; Jiang et al, 2017). For example, mutations of the gene encoding ligase 4 (*Lig4*) are a cause of microcephaly (Buck et al, 2006), whereas *XRCC1* mutations have been associated with seizures (Hoch et al, 2017). However, the role of PNKP in postnatal brain function still remains poorly understood. Intriguingly, one study linked *Pnkp* mutations to an early onset neurodegenerative disorder (Poulton et al, 2013), although this was not in keeping with an earlier study (Shen et al, 2010). More recently, mutations in *Pnkp* have been linked to a Charcot–Marie–Tooth–like disease ("CMT-like"), with age of disease onset in affected individuals in their twenties (Leal et al, 2018).

To gain insight into the functional role of PNKP in the brain as well as other tissues of adult mice, we used the Cre-*lox*P technology to generate a global *Pnkp* KO using the 4-hydroxytamoxifen (4-OHT)–inducible ubiquitin C (UBC)-CreER$^{T2}$ driver. *Pnkp* deletion was initiated at 3 wk of age, and this led to a progressive age-dependent phenotype in which *Pnkp*-deficient mice exhibited: (i) alopecia resulting from hair follicle (HF) degeneration followed by progressive hyperpigmentation, (ii) a complete absence of spermatogenesis, and (iii) loss of brain subventricular zone neural progenitors and diminished neurogenesis. In each tissue, *Pnkp* deficiency was accompanied by evidence of DNA damage (as shown by γH2AX staining) within active progenitor pools, and this was accompanied by dramatically increased levels of cell death within the respective progenitor populations. Thus, our results indicate that normal PNKP function is critical to the survival of progenitors in diverse murine tissues, including skin, testes, and brain.

# Results

### Generation of mice lacking *Pnkp* in all tissues

Global or developmental CNS loss of *Pnkp* is embryonically lethal (Shimada et al, 2015); thus, to investigate the role of *Pnkp* in postnatal animals we generated mice allowing temporal control over *Pnkp* deletion via the use of the Cre-*lox*P technology. To accomplish this, a *Pnkp*$^{fl/fl}$ transgenic line was developed containing a *lox*P-flanked ("floxed") exon 2. Deletion of this exon is predicted to eliminate the FHA domain of PNKP required for functional protein–protein interactions (Koch et al, 2004; Loizou et al, 2004) (Fig 1A–D). Furthermore, if bypass splicing from exon 1 to exon 3 were to occur as a result of exon 2 deletion, this would be predicted to result in a frame-shift followed by an early termination codon (Fig 1A–D). To attain global KO of *Pnkp* in adult mice we generated UBC-CreER$^{T2}$:*Pnkp*$^{fl/fl}$ mice by crossing our *Pnkp*$^{fl/fl}$ mice with mice expressing the UBC-CreER$^{T2}$ transgene (Fig 1E). This Cre driver is expressed in all mouse tissues (Ruzankina et al, 2007) and its recombinase activity can be trigged by tamoxifen administration. The resulting mice are designated as *Pnkp* KOs herein, whereas control mice with genotype *Pnkp*$^{fl/fl}$ are designated as *Pnkp* WT.

Similar to the Nestin-Cre x *Pnkp*$^{fl/fl}$ deletion experiment reported previously (Shimada et al, 2015), we found that interbreeding the Nestin-Cre driver with our *Pnkp*$^{fl/fl}$ mice resulted in neonatal lethality. Specifically, no viable Nestin-Cre–induced *Pnkp* KO pups were identified after the birth of 54 pups where the predicted KO frequency was 25%. Furthermore, the 54 births included 11 stillborn pups, of which 4 were confirmed to be *Pnkp* KOs by genotyping. Last, in a timed-pregnancy pilot experiment, three of nine fetuses taken at 18.5 d gestation were found to be *Pnkp* KOs. Thus, Nestin-Cre–mediated KO of the floxed *Pnkp* led to pre- and peri-natal mortality.

To trigger Cre-induced deletions of exon 2 (Fig 1F), UBC-CreER$^{T2}$:*Pnkp*$^{fl/fl}$ mice were administered 4-OHT. To confirm loss of PNKP protein in various tissues, we used a rabbit polyclonal anti-PNKP antibody (generated in the laboratory of M Weinfeld) that was raised against the full-length recombinant human PNKP protein, but that also cross-reacts with mouse PNKP. Immunoblotting confirmed the lack of PNKP protein expression in multiple *Pnkp* KO mouse tissues (Fig 1G and H). Immunostaining of skin, brain and testis tissue sections further demonstrated loss of PNKP protein expression (Fig 1I–K).

As detailed below, deletion of *Pnkp* exon 2 led to a number of salient phenotypes, including hair loss, hyperpigmentation, failure to gain weight, diminished dermal fat layer, and a complete absence of spermatogenesis (Fig S1A–H). This striking phenotype was not evident in *Pnkp*$^{fl/fl}$ mice, consistent with these *Pnkp* alleles not being hypomorphic as a result of the insertion of the *lox*P sequences flanking exon 2.

### Loss of *Pnkp* leads to a defective HF regeneration cycle

The mammalian HF cycles through stages of active growth (anagen), degeneration (catagen), and quiescence (telogen) to regenerate hair throughout life. This regenerative process is entirely dependent on resident epithelial and mesenchymal stem/progenitor populations that function to supply new cells to the growing HF (Müller-Röver et al, 2001; Stenn & Paus, 2001; Hsu et al, 2014; Morgan, 2014). Within several weeks of *Pnkp* deletion, mice exhibited hair thinning and by 3 mo marked body hair loss was observed (Figs 1L and M and S1D). To determine whether PNKP plays a role in adult HF-regenerative cycle, age-matched WT and *Pnkp* KO mice were depilated (plucked) at 2, 5, and 8 mo of age and hair regrowth was then monitored (Fig 2A). Regardless of age, all *Pnkp* KO mice failed

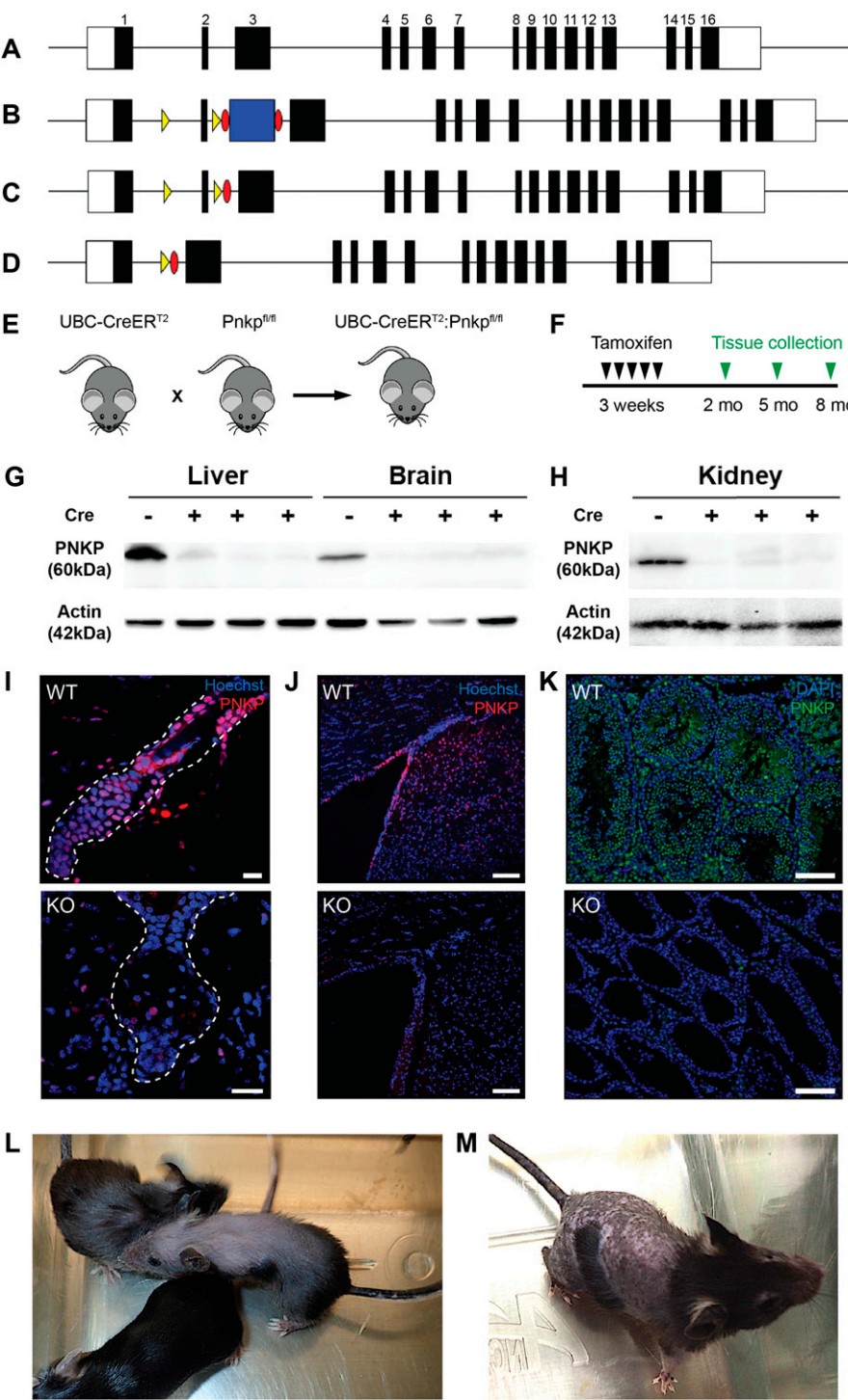

**Figure 1. Generation of UBC-CreER^T2:*Pnkp^fl/fl* mice.**
**(A)** Diagram of the *Pnkp* genomic locus showing the exon–intron organization. The white boxes represent the 5'- and 3'-untranslated regions and black boxes represent each labelled exon. **(B)** Diagram of the *Pnkp* locus after insertion of *lox*P sites (yellow arrowheads) flanking exon 2, as well as the Flp recombinase recognition targets (*Frt*) (red ovals) flanking the *Pgk*-Neo^R selection cassette (blue box). **(C)** Diagram of the *Pnkp* locus after deletion of the selection cassette after a cross with the Flp recombinase–expressing transgenic line. **(D)** Diagram of the Cre-mediated deletion of the *lox*P-flanked exon 2. **(E)** Crossing of the UBC-CreER^T2 transgenic line with the *Pnkp^fl/fl* mice generated the UBC-CreER^T2:*Pnkp^fl/fl* mice (referred to herein as *Pnkp* KO mice) used in the experiments. **(F)** Timeline for tamoxifen treatment and tissue analysis. **(G, H)** Representative immunoblot showing PNKP protein (60 kD) and Actin (42 kD) expression in the liver, brain, and kidney in *Pnkp^fl/fl* mice and the loss of PNKP expression following tamoxifen (4-OHT)-treatment of three UBC-CreER^T2:*Pnkp^fl/fl* mice (Cre+ lanes). Mice were treated with 4-OHT at 3 wk of age and euthanized at week 12. Lanes were loaded with 50 µg of protein from each tissue and after transfer, the membranes were immunoblotted with the anti-PNKP R626 rabbit polyclonal antibody. **(I, J, K)** Representative immunohistochemistry images for the PNKP protein in (I) hair follicle of 8 mo old UBC-CreER^T2:*Pnkp^fl/fl* mice (red; scale bars = 15 µm), (J) brain of 5 mo old UBC-CreER^T2:*Pnkp^fl/fl* mice (red; scale bars = 100 µm), and (K) testis of UBC-CreER^T2:*Pnkp^fl/fl* mice (green; scale bars = 100 µm). **(L, M)** Representative images of 8 mo old UBC-CreER^T2:*Pnkp^fl/fl* mouse treated with 4-OHT at 3 wk. **(L, M)** Alopecia and hair greying were observed as well as (M) abnormal hyperpigmentation of the denuded skin.

to regrow hair after back skin depilation (n = 5/group/age; Fig 2A). Cross-sectional analysis of *Pnkp* KO mice skin revealed that HFs underwent a stepwise degeneration, reminiscent of that described for age-related degeneration (Fig 2B) (Matsumura et al, 2016). Representative images of each degenerative stage are shown for *Pnkp* KO mice. Step 1 is represented by degeneration of the HF

mesenchyme, known as the dermal papilla (DP; Fig 2B). Step 2 shows the complete detachment or loss of the DP (Fig 2B), and Step 3 shows the degenerating epithelium, resulting in an atrophic overall HF structure (Fig 2B). The age-related stepwise HF degeneration is thought to be due to progressive stem/progenitor cell loss within the regenerative mini-organ (Matsumura et al, 2016).

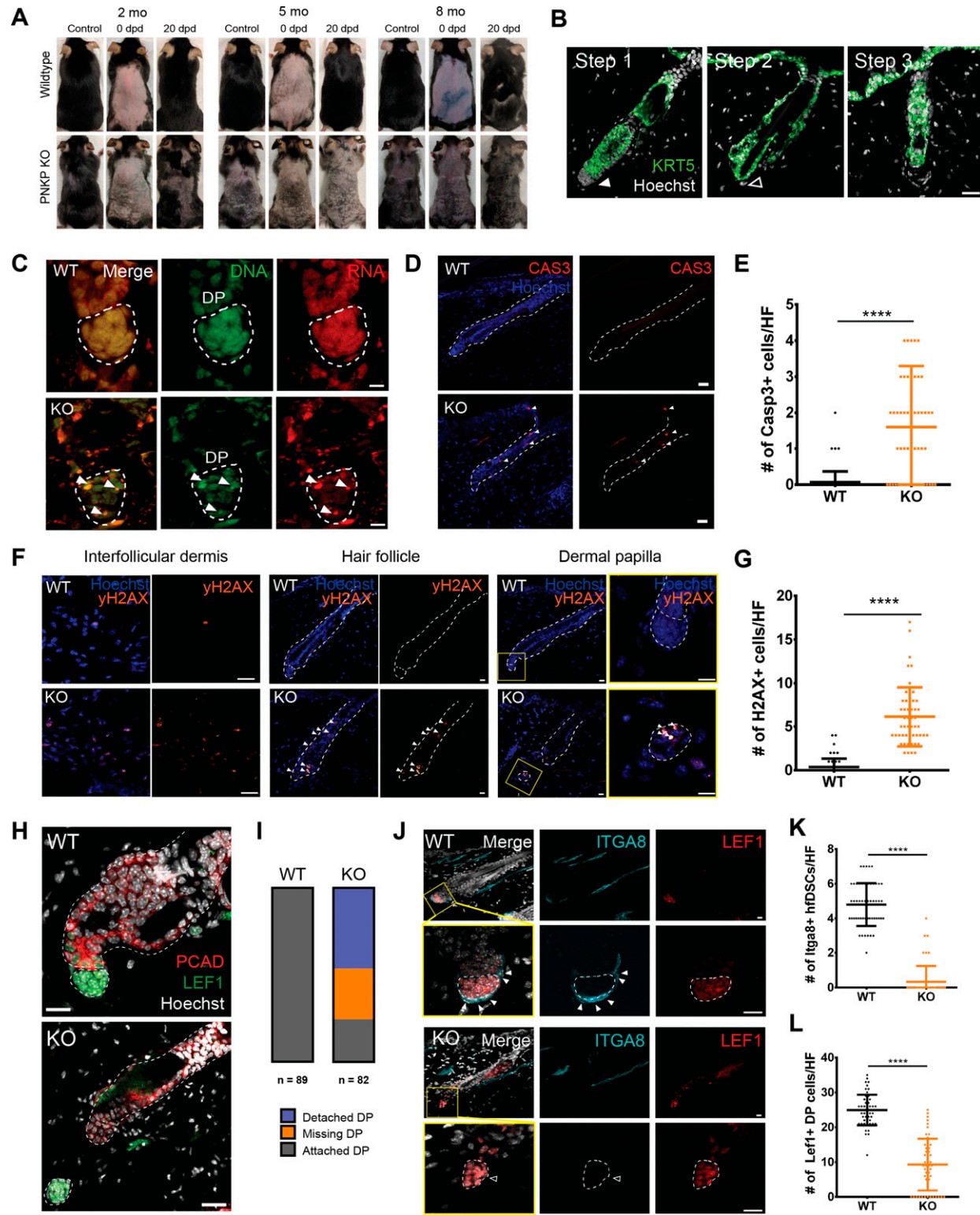

**Figure 2.** *Pnkp* deficiency leads to progressive hair loss and degeneration of the hair follicle (HF) mesenchyme and progenitor cells.
**(A)** Representative images of hair regrowth of tamoxifen-treated UBC-CreER[T2]:*Pnkp*[fl/fl] mice after depilation at 2, 5, and 8 mo of age. Images were taken before depilation (control), immediately after depilation (0 dpd) and 20 d post-depilation (20 dpd). **(B)** Images of 5 mo old *Pnkp* KO skin stained with KRT5, characterizing the stepwise degeneration of HFs after the loss of *Pnkp*. Step 1 represents an intact HF. The filled white arrow indicates the presence of mesenchyme. Step 2 represents the degeneration of the HF mesenchyme. The HF devoid of the mesenchyme is highlighted with an empty white arrow. Step 3 represents the miniaturization of the HF epithelium. Scale bar = 25 μm. **(C)** 5 mo old WT and *Pnkp* KO skin stained with acridine orange, staining DNA in green and RNA in red. Bodies with DNA and RNA overlap are highlighted by white arrows, likely indicative of apoptotic debris. **(D)** 5 mo old WT and *Pnkp* KO skin stained with CASP3 (red). Scale bar = 50 μm. **(E)** Quantification of the

### *Pnkp* KO mice accumulate DSBs and show apoptosis within the HF

To investigate whether cells within the HF were undergoing apoptosis or necroptosis, acridine orange staining was used. At a concentration of 50 μM, acridine orange complexes with DNA to emit green wavelengths (~530 nm) and RNA to emit red wavelengths (~630 nm) (Fig 2C) (Plemel et al, 2017). In healthy cells, the RNA signal is found within the cytoplasm and nucleus, whereas the DNA signal is restricted to the nucleus. The presence of cells with normal DNA and RNA distribution and signals in *Pnkp* KO mice suggested that HF degeneration was due to apoptosis rather than to necrosis. We then immunostained WT and *Pnkp* KO skin sections for cleaved Caspase 3 (CAS3) to detect apoptotic cells, and with γH2AX to detect DSBs. The *Pnkp* KO mice showed significant increases in the number of CAS3$^{+ve}$ apoptotic cells within the HF (Fig 2D and E), and there was evidence of DSB throughout the HF and interfollicular dermis (Fig 2F and G). Thus, loss of *Pnkp* resulted in an accumulation of DNA damage within the cells of the HF, and this may have caused apoptosis. Interestingly, cell death was restricted to the HF because CAS3$^{+ve}$ cells were rarely found in the interfollicular dermis (Fig 2D). These results suggested that *Pnkp* expression was essential to the maintenance of HF progenitor populations.

HFs from *Pnkp* KO mice often exhibited either an absence of the DP, or DP detachment and displacement from the epithelial compartment. In the *Pnkp* KO mice, lymphoid enhancer–binding factor 1 (LEF1)$^{+ve}$ DP were observed in ectopic positions in the interfollicular dermis, distant from adjacent HFs (Fig 2H). The DP is a crucial signaling center that modulates transitions in and out of HF regeneration stages. The temporal release of growth factors, including Wnts, transforming growth factors, bone morphogenetic proteins, and fibroblast growth factors, from the DP act to control cell proliferation and differentiation within the HF (Rendl et al, 2008; Greco et al, 2009; Oshimori & Fuchs, 2012). Quantification showed that while 2–5-mo-old WT mice maintained a normal HF structure in all cases (Fig 2I; n = 89 HFs from four mice), the DP was detached in 46.3% of HFs from sex- and age-matched *Pnkp* KO mice, and a further 29.2% of HFs showed a complete absence of DP (Fig 2I; n = 82 HFs from five mice).

Regeneration of the HF mesenchyme (including the DP) is dependent on mesenchymal progenitors known as HF dermal stem cells (hfDSCs) (Rahmani et al, 2014; González et al, 2017). During telogen phase of the HF regeneration cycle, Integrin α8$^{+ve}$ (ITGA8$^{+ve}$) hfDSCs can be distinguished from LEF1$^{+ve}$ DP cells (Fig 2J). Within 3 wk after tamoxifen injection, HFs from *Pnkp* KO mice showed a complete loss of ITGA8$^{+ve}$ hfDSCs (Fig 2J and K). As well, the total number of DP cells that are continually supplemented by hfDSC progeny (Shin et al, 2020) was also diminished (Fig 2B, J, and L; Step 2). The persistence of DP cells in the absence of hfDSCs suggested that *Pnkp* deficiency first impacts the mesenchymal progenitor pool

(hfDSCs), whereas the loss of differentiated DP cells occurs as a secondary event (Fig 2J). Together, our data suggest that PNKP is critical for the maintenance of progenitor cells within the HF, with PNKP deficiency leading to HF degeneration, loss of regenerative capacity, and hence, progressive hair loss.

Next, to investigate whether the loss of *Pnkp* exclusively in mesenchymal progenitors residing in hair follicles can recapitulate similar phenotypes observed in global *Pnkp* KO mice, we generated alpha-Smooth muscle actin-CreER$^{T2}$:Rosa$^{eYFP}$:*Pnkp*$^{fl/fl}$ transgenic mice (herein referred to as αSMA:*Pnkp* KO) (Figure S2A). In skin, αSMACreER$^{T2}$: Rosa$^{eYFP}$ effectively labels dermal sheath progenitors that maintain the inductive mesenchyme within hair follicles, as well as vascular smooth muscle cells and arrector pili muscle (Rahmani et al, 2014; González et al, 2017). 4-OHT was administered at the onset of first and second postnatal HF regeneration (Fig S2B) and the mice were tracked until 3 mo (N = 9) or 12 mo (N = 14) of age. Additionally, an identical 4-OHT treatment regime was applied as done to the UBC:*Pnkp* KO mice (Fig S2C; N = 7). In all experiments, none of the mice exhibited hair loss or hyperpigmentation when examined at 3, 8, or 12 mo of age (Fig S2D and E). Immunofluorescence staining showed YFP+ cells were devoid of the PNKP protein 2 mo after the 4-OHT treatment (Fig S2F), suggesting that self-renewing mesenchymal progenitors can survive in their niche in vivo without functional *Pnkp* for at least 2 mo. Together, the data suggested that neither 4-OHT administration, nor 4-OHT plus *Pnkp* deletion, was sufficient to cause depletion of fibroblast progenitors, and furthermore, that depletion of mesenchymal progenitors did not appear to play a role in the hair loss phenotype.

### Pigment-containing cells emigrate from the degenerating HF and contribute to ectopic hyperpigmentation within the dermis of *Pnkp* KO mice

Melanin, synthesized by melanocytes, is stored in large pigment organelles called melanosomes (Slominski et al, 2004, 2005), and in response to ultraviolet light–induced DNA damage, keratinocytes produce factors that stimulate melanin synthesis. In mice, the melanocytes at the base of the hair bulb, which provide melanin to growing hair fibers, can be detected with an antibody against the melanosome protein glycoprotein 100 (GP100) (Fig 3A) (Slominski et al, 2004, 2005). Interestingly, in 2-mo-old *Pnkp* KO mice, melanin-bearing cells had expanded into the HF epithelium (Fig 3B; Step 1), whereas at later time points (5–8 mo), the melanin-containing cells had spread throughout the entire follicle (Fig 3B; Step 2), and eventually appeared to be emigrating out of the follicle and into the interfollicular dermis (Fig 3B; Step 3). Small dense melanin deposits were observed in *Pnkp* KO tails (Figs 3C and S1E), whereas much more widespread deposits were found throughout the back dermis

number of CASP3$^{+ve}$ cells/HF in WT and KO mice (n = 3 mice/group). **(F)** Images of 5 mo old WT and *Pnkp* KO skin stained with γH2AX (orange) at the interfollicular dermis, HF and dermal papilla (DP). Scale bar = 15 μm. **(G)** Quantification of the number of γH2AX$^{+ve}$ cells/HF in WT and KO mice (n = 3 mice/group). **(H)** 5 mo old *Pnkp* KO HFs stained with PCAD (red) and LEF1 (green) representing separation of the HF mesenchyme from its epithelium. Scale bar = 15 μm. **(I)** Quantification of the number of HFs with detached DP, missing DP or attached DP. (n = 3 mice/group). **(J)** Number of cells comprising the DP is reduced and HF dermal stem cells (hfDSCs) are lost in *Pnkp* KO mice; 2 mo old *Pnkp* KO mice tissue stained with ITGA8 (cyan) to label hfDSCs and LEF1 (red) to label DP in telogen HFs. Scale bar = 15 μm. **(K)** Quantification of the number of LEF1$^{+ve}$ DP cells per HF (n = 3). **(L)** Quantification of the number of ITGA8$^{+ve}$ hfDSCs per HF (n = 3). **(E, G, K, L)** In (E, G, K, L), data are presented as mean ± SD. ****$P$ > 0.001 (two-sided $t$ test).

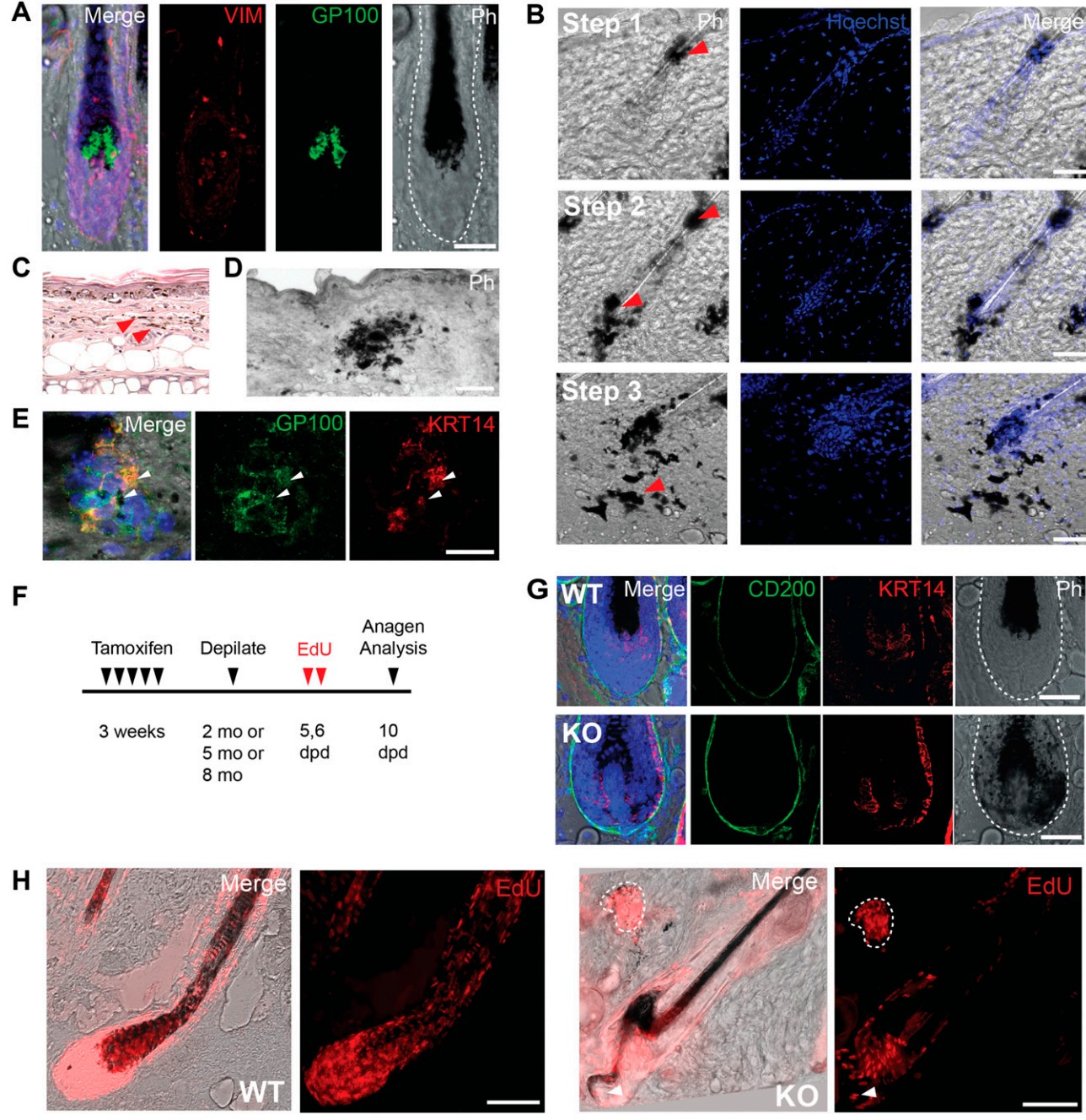

**Figure 3.** ***Pnkp* deficiency drives expansion of melanin-bearing cells throughout the murine dermis.**
**(A)** 5 mo old WT hair follicle (HF) stained with Vim (red) and Gp100 (green). Gp100[+ve] melanocytes are localized to the base of the hair shaft. Scale bar = 50 μm.
**(B)** Stepwise expansion of melanin expressing cells derived from within the HF of 8 mo old *Pnkp* KO mice. Step1: Melanin-containing cells are found at a small region within the HF. Step 2: Melanin-containing cells are spread throughout the entire HF, yet are maintained within the HF. Step 3a: Melanin-containing cells move out into the interfollicular dermis in the *Pnkp* KO. Scale bar = 50 μm. **(C)** H&E staining of a 6 mo old *Pnkp* KO mouse tail. Melanin is spread all throughout the dermis (100×).
**(D)** Representative phase contrast image of 8 mo old *Pnkp* KO mouse back skin. Melanin is shown in black. Scale bar = 100 μm. **(E)** Dermal melanin deposits in 8 mo old *Pnkp* KO mouse back skin stained with GP100 (green) and KRT14 (red). Scale bar = 15 μm. **(F)** Experimental timeline of 4-OHT and EdU injections to track proliferating cells in *Pnkp* KO mice. Depilation induces synchronized hair regeneration and EdU was administered 5–6 d post-depilation to label proliferating hair follicle cells.
**(G)** Anagen HFs from 8 mo old *Pnkp* KO mice labelled with EdU. Melanin clusters contain EdU[+ve] cells outlined with white dotted line. Arrow indicates melanin positive cells escaping the HF during anagen. Scale bar = 100 μm. **(H)** Representative images of the HF in WT (top) and *Pnkp* KO (bottom) 5 mo old back skin stained with CD200 (green) and KRT14 (red). Melanin can be found in KRT14[+ve] matrix cells in the *Pnkp* KO, whereas melanin is contained exclusively within the hair fiber in WT animals. Scale bar = 50 μm.

of 5–8 mo mice (Figs 1M and 3D). Cells within the latter deposits, found throughout the interfollicular dermis, were positive for GP100 and the keratinocyte marker keratin 14 (KRT14; Fig 3E). This suggested that both melanocytes and keratinocytes were escaping the HF boundary and generating compound aggregates within the *Pnkp* KO dermis.

To confirm that GP100[+ve] cells were originating from the HF, we performed an EdU pulse-chase experiment after inducing synchronized hair growth. 2–8 mo-old WT and *Pnkp* KO mice were depilated to induce entry into anagen, and EdU was then injected i.p. 5 and 6 d after depilation to label proliferating cells. The anagen HFs were then analyzed 10 d post-depilation (Fig 3F). As expected, melanin was restricted to the growing hair fiber during anagen generation in WT mice. However, in *Pnkp* KO mice, KRT14[+ve] epithelial cells of the HF were absorbing the melanin as well (Fig 3G). Interestingly, CD200[+ve] connective tissue sheath cells, along with DP cells were devoid of melanin absorption (Fig 3G). In addition, oddly shaped anagen HF were observed with EdU[+ve] cells moving out of the HF, suggesting that the structural integrity of the HFs was disrupted. EdU[+ve] cells were also observed in melanin deposits in the interfollicular dermis (Fig 3H). These findings suggested that after loss of *Pnkp*, melanocytes and pigmented keratinocytes had migrated away from the follicle structure with retention of their proliferative capacity within the interfollicular dermis.

### *Pnkp* KO testes show a dramatic impairment of spermatogenesis

Given the finding that *Pnkp* KO testes lacked spermatozoa (Figs 4A and S1F and G), spermatogenesis in these mice was investigated. Spermatogonial stem cells, the initiating cells in spermatogenesis, undergo division to generate progenitor spermatogonia which amplify and subsequently undergo two meiotic divisions to yield spermatids; the latter then repackage their DNA from histones to protamines, undergo considerable morphological changes, and form haploid spermatozoa. Furthermore, spermatogenesis occurs in the long, continuous seminiferous tubules that make up the testis and is supported and regulated by Sertoli cells, which also reside in the tubules, as well as Leydig and other cells types that make up the interstitial space between the tubules (Oatley & Brinster, 2006). To investigate whether this process was impacted in the *Pnkp* KO mice, testes were collected from mice at 24 h, 7 d, 1, and 5 mo post–4-OHT administration. To avoid 4-OHT-induced DNA damage to spermatogonia at extreme high doses (Sadeghi et al, 2019), 4-OHT treatment was limited to 5 d at 75 mg/kg. Whereas the histology of *Pnkp* KO testes appeared normal 24 h post–4-OHT, after 1 wk, the seminiferous tubules exhibited disruption, with the remaining germ cells detaching from the seminiferous epithelium (Fig 4A). By 2 mo of age, *Pnkp* KO seminiferous tubules were devoid of spermatogenesis, and by 5 mo, the tubular architecture was significantly disrupted (Fig 4A), and eventually the testes became atrophic (Fig S1F).

### Undifferentiated spermatogonia proliferate but do not differentiate in *Pnkp* KO mice

Immunostaining of cell type–specific markers within the testis was used to further characterize the spermatogenesis defect in *Pnkp* KO

mice. We observed a few persisting Lin 28 homolog A (LIN28A)[+ve] germ cells that represent progenitor spermatogonia (Chakraborty et al, 2014); however, no further differentiating germ cells were observed as shown by the absence of meiotic synaptonemal complex protein 3–positive (SYCP3)[+ve] germ cells (Fig 4B) (Yuan et al, 2000). These rare, persistent germ cells were positive for both glial cell line–derived neurotrophic factor receptor α 1 (GFRα1) and the promyelocytic leukaemia zinc finger protein (PLZF) that are established protein markers for undifferentiated spermatogonia (Fig 4C and D) (Buaas et al, 2004; Naughton et al, 2006). These undifferentiated spermatogonia were mitotically active, evident by proliferating cell nuclear antigen expression. We also examined the presence of somatic Sertoli and Leydig cells, which comprise key functional components of the spermatogonial stem cell niche. Vimentin (Vim)-expressing Sertoli, as well as Leydig cells were present in the *Pnkp* KO testes (Fig 4E). By 5 mo of age, the tubules, which now predominantly comprised Sertoli cells, contained only the occasional cluster of spermatogonia. Thus, similar to our findings in the HF, loss of *Pnkp* negatively impacts progenitor expansion in the murine testes.

### Increased DNA damage and apoptosis in spermatogonial progenitors

To further examine the impact of *Pnkp* loss in spermatogonia, cultures of these cells, isolated from UBC-CreER[T2]:*Pnkp*[fl/fl] testes, were expanded in vitro. Thus, before treatment with 4-OHT, spermatogonia were isolated from 6–8 d WT and UBC-CreER[T2]:*Pnkp*[fl/fl] mice and cultured for five passages to obtain sufficient cell numbers for experimentation. We found that UBC-CreER[T2]:*Pnkp*[fl/fl] spermatogonia contained significantly fewer cells by 6 d post 4-OHT treatment, as compared with WT 4-OHT–treated or to non-4–OHT treated UBC-CreER[T2]:*Pnkp*[fl/fl] spermatogonia (Fig 4F and G). This decrease in UBC-CreER[T2]:*Pnkp*[fl/fl] cell numbers post-tamoxifen was accompanied by an increased levels of DNA damage (as assessed by γH2AX staining) and the apoptosis marker, cleaved Casp3, despite there being no significant change in their proliferation (Fig 4H).

Next, qPCR was used to assess mRNA expression levels of specific genes in the UBC-CreER[T2]:*Pnkp*[fl/fl] spermatogonia cultures after 4-OHT treatment. *Pnkp* mRNA was significantly reduced, consistent with Cre-mediated recombination having occured in vitro. Interestingly, there is a higher relative gene expression of undifferentiated spermatogonia associated *Id4* in 4-OHT treated UBC-CreER[T2]:*Pnkp*[fl/fl] spermatogonia cultures as compared with tamoxifen-untreated, and tamoxifen-treated WT spermatogonia cultures (Fig 4I). An additional 5 d in culture post–4-OHT led to even further reductions in *Pnkp* KO spermatogonial cell numbers, and usual colony architecture of the cultures was no longer evident (Fig 4J) along with persistence of the relatively high expression of the undifferentiated spermatogonia associated *Id4* gene (Fig 4K). In keeping with the in vivo *Pnkp* KO testis phenotype of massively impaired spermatogenesis, we found that spermatogonia progenitors in culture rapidly accumulated DNA damage and developed evidence of apoptosis following 4-OHT–triggered induction of *Pnkp* deletion. Furthermore, treatment of spermatogonial cells with tamoxifen did not lead to an increase in oxidative stress (Fig 4L).

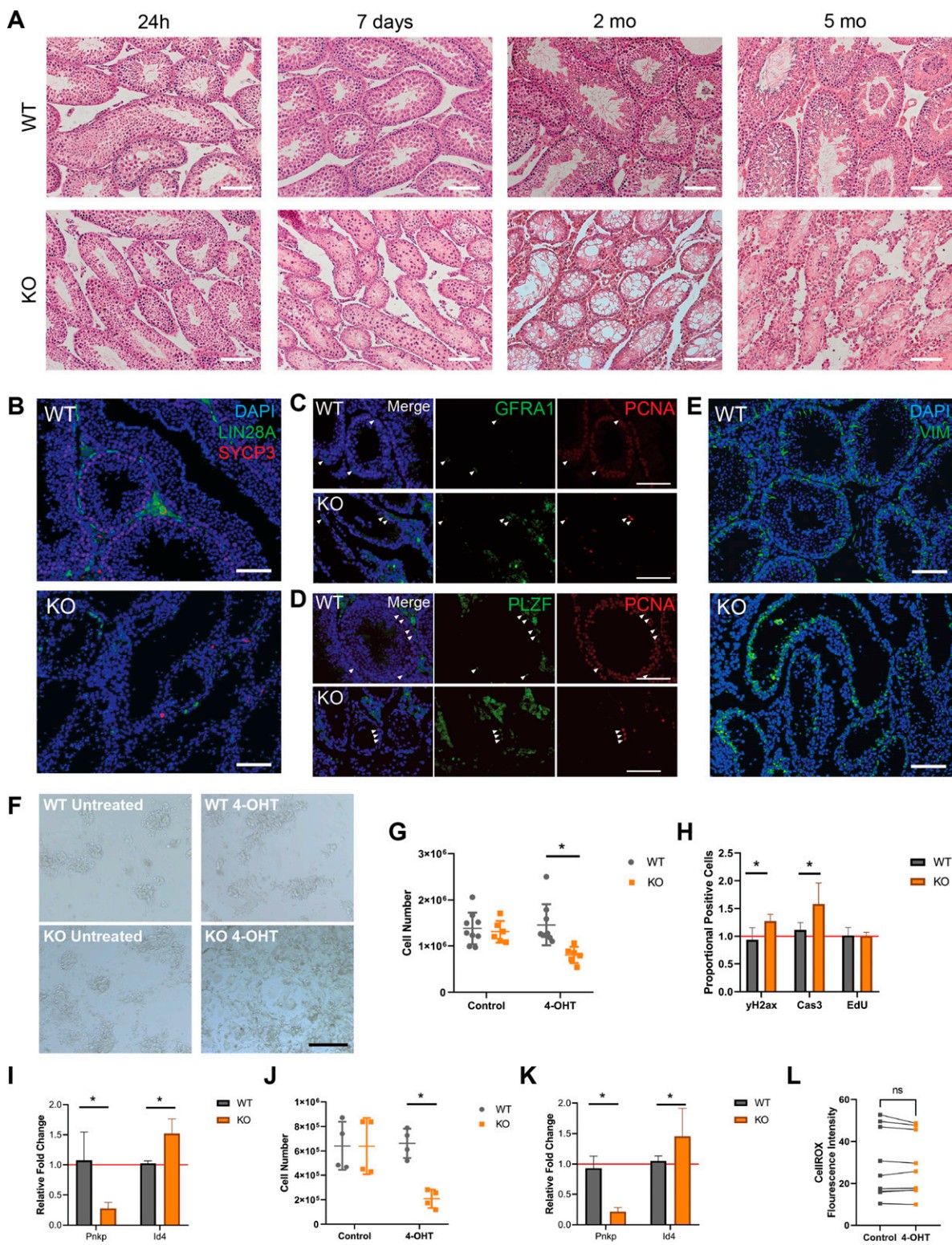

**Figure 4.  *Pnkp*-deficient testes are devoid of spermatogenesis and lack differentiating germ cells.**
**(A)** WT and *Pnkp* KO testis tissue histology with H&E staining 24 h, 7 d, 2, and 5 mo post 4-OHT treatment. Scale bars = 100 μm. **(B)** WT and KO testis tissue 20 wk post 4-OHT stained with DAPI (blue), LIN28a (green), and SYCP3 (red). Scale bar = 100 μm. **(C)** *Pnkp* mutant testis retain proliferating, undifferentiated spermatogonia. WT and *Pnkp* KO testis tissue 8 wk post 4-OHT stained for DAPI (blue), GFRA1 (green), and proliferating cell nuclear antigen (PCNA) (red). White arrows indicate cells double-positive for GFRA1 and PCNA. Scale bar = 100 μm. **(D)** WT and *Pnkp* KO testis tissue 8 wk post 4-OHT stained for DAPI (blue), PLZF (green), and PCNA (red). White arrows indicate cells double positive for PLZF and PCNA. Scale bar = 100 μm. **(E)** WT and *Pnkp* KO testis tissue 3 wk post 4-OHT stained for DAPI (blue) and Vim (green). Scale bar = 100 μm.
**(F)** Representative phase contrast images of cultured spermatogonia for 6 d from WT mice (WT untreated), WT mice treated with tamoxifen (WT 4-OHT), *Pnkp* KO mice

## Deletion of *Pnkp* leads to increased DNA damage within cells within the subventricular zone (SVZ) accompanied by depletion of neural stem cells (NSCs) and their progeny

Prior studies have shown that *PNKP* mutations lead to neurological abnormalities such as MCSZ in humans, and greatly impaired neurodevelopment in mice, and that *Pnkp* mRNA is expressed within subventricular progenitor cell populations of both humans and mice (Shen et al, 2010). Based on this, and given the defects observed in response to *Pnkp*-deficiency in progenitors of skin and testes, we hypothesized that adult neurogenesis in the subventricular zone of the lateral ventricle (V-SVZ) might also be negatively impacted by *Pnkp* loss.

Postnatal NSCs exist as both quiescent NSCs (qNSCs) and active NSC states (aNSCs). The aNSCs give rise to doublecortin+ve (DCX+ve) neuroblasts that populate the olfactory bulb with new interneurons via the rostal migratory stream. To explore the consequences of *Pnkp* loss on these cells, both WT and UBC-CreER^T2:*Pnkp*^fl/fl mice were treated with 4-OHT at 3 wk of age and then administered EdU for 2 d when they reached 2–8 mo of age (Fig 5A). Immunostaining for γH2AX revealed a marked increase in the number of positive cells in *Pnkp* KO, versus WT brains (Fig 5B). Evaluation of DCX+ve neuroblasts showed that the number of DCX+ve neuroblasts in the dorsolateral horn was significantly decreased in *Pnkp* KO brains (Fig 5C and D). Administration of EdU revealed that *Pnkp* loss resulted in a significant reduction in the numbers of EdU+ve cells per ventricle, as compared with 4-OHT–treated WT mice (Fig 5C and E). To evaluate neural progenitors, we quantified the total number of GFAP+ve SOX2+ve cells (i.e., cells co-expressing GFAP and the SOX2 transcription factor that are essential for maintaining NSCs), along the dorsal horn, lateral wall, and tail of the SVZ from each ventricle in WT and *Pnkp* KO animals. Again, *Pnkp* KO brains demonstrated significant reductions in the numbers of GFAP+ve SOX2+ve cells per ventricle (Fig 5F and G). These results support the notion that PNKP deficiency dramatically disrupts NSC maintenance, thus limiting adult neurogenesis.

### *Pnkp* deletion in neural progenitors results in diminished capacity for self-renewal

To evaluate how *Pnkp* deletion might impact the self-renewal capacity of neural progenitors in vitro, clonally derived neurospheres were generated from the brains of 6–8 d old WT and UBC-CreER^T2:*Pnkp*^fl/fl mice. These were passaged twice to enrich for self-renewing stem/progenitors, and then treated with 4-OHT in vitro for 2 d (Fig 5H). Analysis 9 d later revealed that neurospheres grown from *Pnkp* KO mice were not only significantly smaller in size, but the frequency of spheres per well was also reduced in comparison to 4-OHT–treated WT control cells (Fig 5I–K). Depletion of *Pnkp*

mRNA in the cells of *Pnkp* KO cultures was confirmed by qPCR (Fig 5L). Congruent with the effects of *Pnkp* loss on spermatogonial progenitors, we found that neural progenitors were defective in self-renewal and proliferation after induction of *Pnkp* deletion.

### CRISPR-Cas9 induced deletion of *PNKP* failed to sensitize cultured cells to 4-OHT

.Whether or not loss of *PNKP* sensitizes cells to 4-OHT induced cytotoxicity remained an important question. To investigate this, *PNKP* was deleted in the HCT116 colorectal cancer cell line via use of CRISPR-Cas9 (Fig S3A). Control and *PNKP* KO HCT116 cells were then exposed to increasing concentrations of 4-OHT (including concentrations much higher than those we used in the in vitro experiments on neural and spermatogonial cells) so as to evaluate the potential effect of this agent on HCT116 growth and survival. Crystal violet-based viability assay showed that treatment with 4-OHT did not lead to a preferential decline in the growth of *PNKP* KO cells, as compared to controls (Fig S3B). Additionally, clonogenic colony forming assays demonstrated that *PNKP* loss failed to sensitize HCT116 cells to 4-OHT (Fig S3C). In contrast, and as shown by the colony forming assay, *PNKP* deficiency sensitized the cells to ionizing radiation, an oxidative stress inducer (Fig S3D). These results provided additional support for the idea that *PNKP* loss does not sensitize cells to potential 4-OHT-induced cytotoxicity.

## Discussion

Our results highlight the importance of PNKP in maintaining tissue homeostasis of the post-natal mouse skin, testis, and brain. Indeed, proliferative progenitor cells may be highly susceptible to genotoxic stress and as a result rapidly undergo apoptosis if defective in DNA repair proteins such PNKP. We speculate that this accounts for the impairments in HF cycling, spermatogenesis, and neurogenesis that we have observed. Furthermore, our results suggest that postnatal induction of *Pnkp* deficiency leads to a progressive degenerative phenotype akin to premature aging. Indeed, stem cell exhaustion or depletion is thought to be one of the factors contributing to the aging process (López-Otín et al, 2013; Vermeij & Hoeijmakers, 2016). Clearly, aging can also be accelerated by the accumulation of nuclear and/or mitochondrial DNA damage, with the latter also having the potential to contribute to the aging process via senescence induction (Vijg & Suh, 2013; Vermeij & Hoeijmakers, 2016).

DNA damage in response to reactive oxygen species (ROS), such as hydrogen peroxide, typically leads to single strand DNA breaks (SSBs), and more rarely to double strand DNA breaks (DSBs). In either scenario, 3'-PO$_4$ and 5'-OH can result from this type of

---

(KO untreated), and *Pnkp* KO mice treated with 4-OHT. Scale bar = 100 μm. **(G)** Quantifications of spermatogonia cell numbers after culture at 6 d. **(H)** Flow cytometry counts for CASP3, γH2AX, and EdU presented as proportion of positive cells in the 4-OHT treatment group relative to the control, untreated, cells represented by red line. **(I)** Relative gene expression of *Pnkp* and *Id4* normalized to *Hprt1* at 6 d. Represented as proportional expression of 4-OHT–treated cultures relative to untreated cultures (set to 1 as indicated by red line). **(J)** Quantifications of spermatogonia cell numbers after culture for 11 d. **(K)** Relative gene expression of *Pnkp* and *Id4* normalized to *Hprt1* at 11 d. Represented as proportional expression of 4-OHT treated cultures relative to untreated cultures (set to 1 as indicated by red line). **(G, H, I, J, K)** In (G, H, I, J, K), data are presented as mean ± SD. *P < 0.05 (two-sided *t* test). **(L)** Oxidative stress measured by median fluorescence intensity of CellROX Green Reagent in cultured WT untreated (Control; grey) or WT 4-OHT–treated (4-OHT; orange) treated spermatogonia at 6 d in culture.

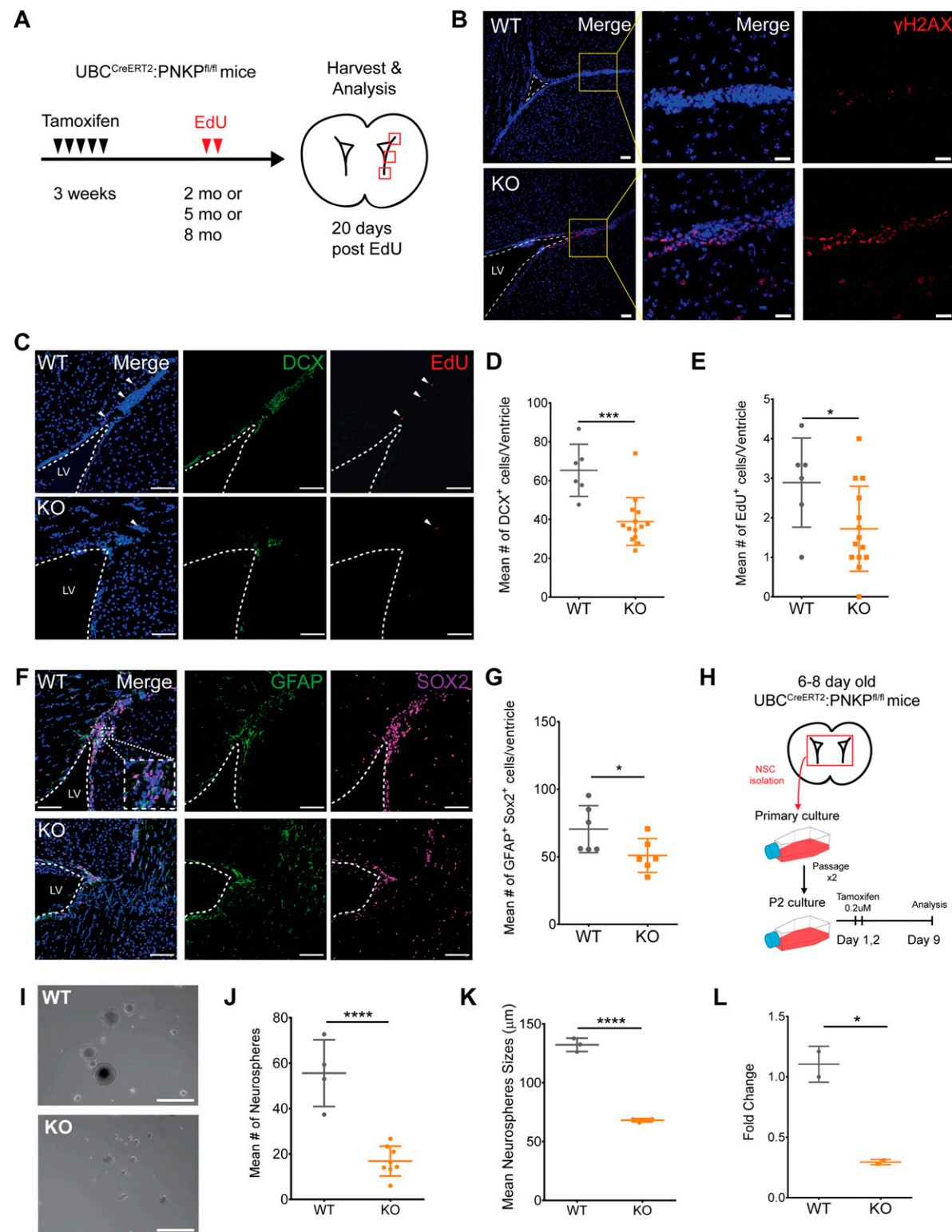

**Figure 5. Loss of *Pnkp* leads to increased DNA damage within the SVZ, subsequently leading to loss of neuroblasts and slow-cycling NSCs.**
**(A)** Experimental timeline of tamoxifen treatment at 3 wk of age, followed by EdU injections, and tissue harvest and analysis at 2, 5, or 8 mo of age. **(B)** 5-mo-old WT and *Pnkp* KO subventricular zones (SVZ) stained with Hoechst (blue) and for γH2AX (red). Scale bar = 50 μm. **(C)** 2-mo-old WT and *Pnkp* KO SVZ stained with Hoechst (blue), DCX (green) to label neuroblasts, and EdU (red) to label slow-cycling neural stem cells (NSCs). White arrows indicate EdU[+ve] slow-cycling NSCs. Scale bar = 100 μm. **(D)** Mean number of DCX[+ve] cells per ventricle for WT (*n* = 3) and *Pnkp* KO (*n* = 7) mice. (*P* = 0.002). **(E)** Mean number of EdU[+ve] cells per ventricle for WT (*n* = 3) and *Pnkp* KO (*n* = 7) mice. (*P* = 0.041). **(F)** WT and *Pnkp* KO SVZ stained for Hoechst (blue), glial fibrillary acidic protein (green) and SOX2 (purple) to label NSCs. Scale bar = 100 μm. **(G)** Mean number of double glial fibrillary acidic protein[+ve] SOX2[+ve] cells per ventricle for WT (*n* = 3) and *Pnkp* KO (*n* = 3) mice. (*P* = 0.049). **(H)** Schematic timeline of neurosphere

damage, and represent targets for PNKP-mediated repair; hence, loss of PNKP greatly sensitizes cells to ROS-induced DNA damage (Jilani et al, 1999; Rasouli-Nia et al, 2004). Similarly, Epstein–Barr virus–transformed lymphocytes derived from MCSZ individuals showing a substantial decrease in their PNKP protein expression exhibited increased sensitivity to hydrogen peroxide (Shen et al, 2010; Reynolds et al, 2012). Although quiescent stem cells show low intracellular ROS concentrations, the levels of ROS increase and are subject to tight regulation during normal progenitor cell activation, proliferation, and differentiation (Bigarella et al, 2014). Because the repair of ROS-mediated DNA damage is a key function of PNKP, we hypothesize that the progenitor defects we have observed both in vivo and in vitro are due effects of these DNA damaging agents. The γH2AX staining that we have seen in the *Pnkp* KO cell populations could be the result of persistent ROS-induced dsDNA breaks that result in 3′-PO$_4$ or 5′-OH ends that require repair by PNKP before efficient NHEJ. In addition, inability to correct ROS-induced ssDNA breaks with abnormal termini, would lead to collapse of replication forks during DNA synthesis, triggering apoptosis. Early stages of the latter could also account for γH2AX nuclear staining. It must also be noted that PNKP plays an important role within mitochondria. Thus, its loss also severely impacts mtDNA damage repair given the ROS-rich environment and lack of mtDNA protection by histones (Tahbaz et al, 2012; Bermúdez-Guzmán & Leal, 2019). Defective high-fidelity mtDNA repair could over time lead to increasing mitochondrial dysfunction that would enhance the levels of cellular ROS, in turn leading to increased nuclear as well as mitochondrial DNA damage. A theoretical caveat to explaining the phenotype of the *Pnkp* KO could be that 4-OHT treatment is known to generate tamoxifen-DNA adducts that would require engagement of DNA repair mechanisms, including nucleotide excision repair (Shibutani et al, 2000; McLuckie et al, 2005), yet the DNA lesions generated in the adduct repair process would not be predicted to be PNKP substrates. More plausibly in view of PNKP function, tamoxifen treatment of breast cancer cells can be associated with induction of oxidative stress (Bekele et al, 2016). However, and as detailed in the Results section, and illustrated by Figs S2 and S3, we found no evidence of 4-OHT-induced oxidative stress, or sensitization of *PNKP* KO cells to concentrations of this agent that were considerably higher than those used in our neural and spermatogonial in vitro studies. Furthermore, in vivo 4-OHT administration failed to deplete *Pnkp*-deficient hair follicle mesenchymal progenitors (Fig 2). Together, these results suggest that the phenotype of the *Pnkp*-null mice was unlikely to be due to 4-OHT-induced oxidative stress, although depending on the specific progenitor compartment examined, this agent may have plausibly played a contributory role. However, *Pnkp* deletion produced a profound phenotype in the absence of any exogenous 4-OHT. Specifically, a severe neurodevelopmental phenotype associated with widespread neural progenitor cell death, associated with gH2AX- and TUNEL-positivity, was observed after crossing *Pnkp^{fl/fl}* mice with mice expressing Nestin-Cre or Emx1-Cre (Shimada et al, 2015).

A question raised by our study revolves around the differences in the phenotypes stemming from *PNKP* mutations in humans versus those we observe in *Pnkp* KO mice. To our knowledge, the human disease phenotypes attributed to mutant *PNKP* alleles have been confined to the CNS (Shen et al, 2010; Poulton et al, 2013; Bras et al, 2015; Dumitrache & McKinnon, 2017; McKinnon, 2017; Leal et al, 2018; Gatti et al, 2019), or both CNS and PNS (Leal et al, 2018). Because dermatologic disease and/or reproductive defects have not been reported in humans with *PNKP* mutations, is it possible that the complex multi-tissue *Pnkp* KO phenotype that we have observed is unique to mice lacking the *Pnkp* gene? In addition to species-specific differences, other explanations for the apparent discordance may exist. For example, except for one notable exception (Leal et al, 2018), the neurological disorder that dominates the clinical picture in humans, if often recognized in young individuals. Thus, it is plausible that over an entire human lifespan, additional, non-CNS phenotypes will gradually emerge in humans with *PNKP* mutations. Also, with few exceptions, the disease associated *PNKP* mutations tend to be the result of point mutations rather than mutations that would be equivalent to the *Pnkp* KO. In the case of *PNKP* alleles with small genetic alterations, it is possible that the encoded protein retains some aspect(s) of its function. Indeed, because *PNKP* mutations can result in reduced protein expression, the levels and functionality of the residual PNKP protein may correlate with the characteristic and severity of the observed phenotype (Shen et al, 2010; Poulton et al, 2013; Bermúdez-Guzmán & Leal, 2019). Thus, *PNKP* mutations that lead to a major reduction in PNKP protein levels, or mutations that compromise one or more of the key functions of the encoded protein, may lead to severe rapid-onset developmental defects as seen in MCSZ. The converse may be true, as with *PNKP* mutations that lead to AOA4. It is likely that selection for *PNKP* mutations capable of preserving some level of normal PNKP function are required for embryonic development and post-natal human viability. Consistent with this idea, our results, and the results of others (Shimada et al, 2015), have demonstrated that early *Pnkp* deletion is lethal to mice.

Impaired repair of DNA lesions occurring in neural progenitors, as seen with losses of specific DNA repair molecules, can lead to the activation of DNA damage–sensing mechanisms and triggering of apoptosis, as a way to prevent injured progenitors from further proliferation (Gatz et al, 2011; Mckinnon, 2013; Barazzuol et al, 2017). For example, neural progenitors in *Lig4* mutant mice were shown to undergo apoptosis in response to very low levels of dsDNA breaks (Gatz et al, 2011). Moreover, losses of many of the known SSB or DSB repair molecules have been proposed to lead to DNA damage and apoptosis in neural progenitor pools (Katyal et al, 2007; Breslin & Caldecott, 2009; Enriquez-Rios et al, 2017). Similarly, we found that

---

isolation from intact brains, followed by in vitro tamoxifen treatment and passaging before experimentation. **(I)** Representative image of neurospheres isolated from WT and *Pnkp* KO animals. Scale bar = 200 μm. **(J)** Mean number of neurospheres per well for WT ($n$ = 4) and *Pnkp* KO ($n$ = 8) animals. ($P$ < 0.001). **(K)** Mean neurosphere size for WT ($n$ = 4) and *Pnkp* KO ($n$ = 8) animals. **(L)** Relative gene expression of *Pnkp* after 4-OHT treatment of WT and *Pnkp* KO neurospheres. Represented as proportional expression of 4-OHT–treated cultures relative to untreated cultures (set to 1). **(D, E, G, H, J, K, L)** Data information: In (D, E, G, H, J, K, L), data are presented as mean ± SD. *$P$ < 0.05, ***$P$ < 0.001, ****$P$ < 0.001 (two-sided $t$ test).

loss of *Pnkp* impacts adult mouse neurogenesis; however, these animals, even aged to 12 mo failed to develop any overt evidence of neurological dysfunction. One reason for this discordance is that *PNKP* mutations in humans would be predicted to exert their effects during vulnerable times of embryonic, fetal, and/or post-natal brain development, with this determine the severity of the resulting neurological phenotype. This contrasts with our mouse model wherein *Pnkp* deletions were triggered post-weaning, presumably attenuating the impact of *Pnkp* deficiency on neuro-development. Thus, in contrast to loss of *Pnkp* during early brain development, the neurological phenotype, if any, resulting from impaired neurogenesis in adult *Pnkp* KO mice appears to be subtle. However, it is possible that a neurological defect(s) may emerge with further aging.

We found that adult neurogenesis was greatly impaired in *Pnkp* KO mice, and this was accompanied by in vivo evidence of DNA damage as shown by γH2AX staining of SVZ cell populations. This observation was also supported by our finding that induction of *Pnkp* deletion in NSC cultures caused a rapid inhibition of cell proliferation and self-renewal. In terms of the underlying pathogenetic mechanism(s) stemming from *Pnkp* loss, it seems plausible that our findings in adult mice mirror the neural cell progenitor damage resulting from *Pnkp* deficiency during early development.

We have focused much of the discussion on neural progenitors because neurological impairment is the predominant phenotypic feature of human *PNKP* mutations. However, in the *Pnkp* KO mouse we found two other progenitor cell populations that were also severely compromised, those required for normal HF growth and for spermatogenesis. The discovery that progenitors in these unrelated cell populations were negatively impacted by *Pnkp* loss, raises the possibility that other progenitor populations might be similarly affected, including hematopoietic progenitors and cells involved in regenerating the gut epithelium. If this proved correct, then it would indicate that one of the key functions of PNKP is to maintain the integrity of progenitor cell populations.

# Materials and Methods

## Mouse handling and ethics

Mouse maintenance and experimentation was carried out with ethics approval from the University of Calgary's Animal Care Committee and in accordance with the regulations of the Canadian Council on Animal Care.

## Generation of floxed *Pnkp* mice

Transgenic mice were generated in the University of Calgary's Clara Christie Centre for Mouse Genomics. To introduce *lox*P sites into the *Pnkp* allele, a pair of RNA Zn finger nucleases (ZFN) (designed by Sigma-Aldrich) were used in combination with a targeting donor vector that included an *Frt*-flanked *Pgk*-Neo[R] selection cassette, and a *lox*P-flanked *Pnkp* exon 2 sequence with 800 bp homology arms on either side (based on C57BL/6 sequences). The ZFN cut site was: ACCCCTGTCATCCAGagaatTGGGTGGTAGAGGCATGAG. The

targeting donor vector was synthesized (Celtek), transformed into Stbl2 competent cells (10268019; Thermo Fisher Scientific), grown in Luria broth (244620; BD Difco) with 100 μg/ml Ampicillin (A9518; Sigma-Aldrich), and purified with the PureLink HiPure Plasmid Midiprep Kit (K210015; Thermo Fisher Scientific).

G4 (129 x C57BL/6 F₁) hybrid embryonic stem cells (ES) (George et al, 2007) were transfected by electroporation with the mRNAs encoding the ZFNs along with the donor vector. Targeted cells were placed under G418 selection (15750060; Thermo Fisher Scientific) at a concentration of 275 μg/ml. Surviving clones were picked, expanded, and analyzed by 5′ and 3′ PCR to verify the correct site integration. Positive ES clones were assessed for aneuploidy via metaphase spread preps. Highly (>75%) euploid lines were either aggregated or microinjected into CD1E WT embryos to generate chimeric mice. Both aggregation and micro-injection protocols produced chimeric mice with successful germline transmission of the targeted *Pnkp* allele. Mice showing germline transmission stemming from an aggregation chimera were then bred with the ROSA26-FLPe mouse (JAX Stock No. 009086) to excise the *Pgk*-Neo[R] selection cassette. Mice were then backcrossed for >10 generations on to the C57BL/6J background. Crossing heterozygous mice together produced homozygous "floxed" *Pnkp*[fl/fl] (exon2) mice. After crossing the *Pnkp*[fl/fl] with a Cre driver line, mice were genotyped using the KAPA HotStart Mouse Genotyping Kit (Ref # KK7352; Sigma-Aldrich) and *Pnkp*'HomoF 5′-GACAAGTCCGTGTCTCTAGCAATC 3′ and *Pnkp*3′HomoR 5′-TCCCACCTATTCACTTCGTGTG 3′ primers. The floxed band was 664 bp, the KO band was 247 bp, and the WT band was 551 bp. PCR cycling program was set to 95°C × 3 min (95°C × 15 s, 60°C × 15 s, 72°C × 30 s) for 35 cycles, then 72°C × 5 min, 4°C hold.

## Conditional deletion of the *Pnkp*

To obtain post-natal gene deletions, *Pnkp*[fl/fl] mice were interbred with mice expressing UBC-CreER[T2] (JAX Stock No. 007001 on a C57BL/6J background) to cause *Pnkp* exon 2 loss in all tissues following tamoxifen administration. All experimental and control mice were administered intraperitoneal injections of 75 mg/kg 4-OHT in sunflower oil every 24 h for five consecutive days, beginning at 3 wk of age. The experimental mice were of the genotype UBC-CreER[T2]:*Pnkp*[fl/fl] that were administered 4-OHT, and controls were of the genotypes of UBC-CreER[T2]:*Pnkp*[fl/fl] (minus 4-OHT), WT, and either UBC-CreERT2 (plus 4-OHT) or *Pnkp*[fl/fl] (plus 4-OHT). All mice were weighed at the time of injection to ensure consistent dosing. *Pnkp*[fl/fl] mice were also interbred with transgenic mice that expressed Nestin-Cre (JAX Stock No. 003771 on a C57BL/6J background).

## Immunohistochemistry

### Brain and skin

Animals were euthanized with perfused with 4% PFA in PBS (0.2 mM and pH 7.4) before surgically removing the brain and skin. Removed brains were stored in 4% PFA overnight before being transferred into 30% sucrose in PBS overnight. Back skin biopsies were fixed with 2% PFA in PBS overnight, washed 3× in PBS and treated in 10%, 20%, and 30% sucrose in PBS overnight. Prepared biopsies were snap-frozen in Clear Frozen Section Compound (VWR International). Frozen tissue blocks were sectioned using a Leica 3050s cryostat

## Antibody table.

| Target antigen | Host | Source | Concentration |
|---|---|---|---|
| CASP3 | Rabbit | Abcam | 1:100 |
| CD200 | Rat | Abcam | 1:100 |
| GFRA1 | Goat | Neuromics | 1:100 |
| ITGA8 | Goat | R&D Systems | 1:100 |
| KRT14 | Rabbit | BioLegend | 1:500 |
| KRT5 | Rabbit | BioLegend | 1:500 |
| LEF1 | Rabbit | Abcam | 1:100 |
| LIN28A | Rabbit | Cell Signaling Techologies | 1:400 |
| PCAD | Rat | Abcam | 1:200 |
| Proliferating cell nuclear antigen | Mouse | Dako | 1:400 |
| PLZF | Rabbit | Sigma-Aldrich | 1:400 |
| PNKP | Rabbit | Weinfeld Lab | 1:100 |
| SYCP3 | Mouse | Abcam | 1:100 |
| VIM | Rabbit | Abcam | 1:400 |
| γH2AX | Rabbit | Abcam | 1:500 |

with sections loaded onto Superfrost plus slides (Thermo Fisher Scientific) and stored at −80°C. Whole-mount brains were cut along the rostral-caudal axis in coronal sections at 15 μm thickness. The analyzed sections were contained within a rostral-caudal section of 1,200 μm, beginning at the most caudal point where the SVZ is clearly visible. Back skin samples were cut along the sagittal plane at 45 μm thickness. Frozen tissue sections were rehydrated, then blocked with 10% normal serum containing 0.5% Triton X-100 for 1 h. Primary antibodies were incubated overnight at 4°C, washed 3× with PBS and then incubated with Alexa Fluor secondary antibodies (Invitrogen) at 1:1,000 for 1 h. After 1× PBS wash, sections were stained with 1 mg/ml of Hoechst 33258 (Thermo Fisher Scientific) for 20 min and washed again 3× in PBS before covering with Permafluor (Thermo Fisher Scientific) and a cover slip. High-resolution imaging was performed with the SP8 spectral confocal microscope (Leica). Image editing was performed using Adobe Photoshop, Adobe Illustrator and LASX software (Leica). The list of antibodies used can be found in "Antibodies Table."

### Testes

Testes were fixed in 4% PFA in PBS and paraffin embedded. Immunohistochemistry was performed on 5 μm sections using primary antibodies listed in "Antibodies Table" and species appropriate secondary antibodies.

### In vivo depilation and EdU pulse-chase

*Pnkp* KO mice at 2, 5, and 8 mo of age were depilated using wax strips. To label proliferating cells, 5-ethynyl-2′-deoxyuridine (EdU) in PBS (50 mg/kg) was administered via intraperitoneal injections for 2 d, and an 8-mm skin punch biopsy was harvested 3 d after injection and the brain 20 d after injection. Before tissue collection,

mice were euthanized with an overdose of sodium pentobarbital and transcardially perfused with 20 ml of PBS followed by 20 ml of cold 4% PFA. The dissected brains were post-fixed overnight in 4% PFA at 4°C and allowed to equilibrate overnight in 30% sucrose at 4°C. The brains were then embedded in cryoprotectant (VWR Clear Frozen Section Compound) for storage at −80°C. The SVZ was cut along the rostral-caudal axis in coronal sections (15 μm) using a cryostat (Leica). The analyzed sections were contained within a rostral-caudal section of 1,200 μm, beginning at the most caudal point where the SVZ is clearly visible. The sections were stored at −80°C until further use. EdU staining was performed using a Click-iT EdU Alexa Fluor 555 kit, according to the manufacturer's instructions. Nuclear staining was subsequently performed using Hoechst-33258 (1:100), and the sections were mounted using Permafluor mountant. Sections were imaged for quantification using a Leica SP8 confocal microscope (20× objective, 1.00 μm z-stacks), and representative images were taken using a Leica SP8 confocal microscope (20× objective and 63× objective).

### Neurosphere culture

Whole brains of postnatal day six (P6) UBC-CreER^T2:*Pnkp*^fl/fl mice were isolated and cut in a sagittal direction down the primary sulcus between the two hemispheres to expose the lateral ventricles. Dissection of the lateral ventricles from the brains of WT (n = 4 animals) and *Pnkp* KO (n = 8 animals) was subsequently performed. The lateral ventricles were minced using a scalpel, then dissociated to single cells in 37°C TrypLE (Thermo Fisher Scientific) with trituration using a P200 pipette for no longer than 10 min. Neutralization of the TrypLE occured with the addition of 3:1 DMEM:F12 (Thermo Fisher Scientific). The cells were centrifuged at 210*g* for 5 min and the supernatant removed. Pellets were resuspended in 3:1 DMEM (Thermo Fisher Scientific) to F12 (Thermo Fisher Scientific),

filtered through a 40-µm filter, then centrifuged at 280*g* for 5 min and the supernatant removed. The pellets were resuspended and seeded in growth media (3:1 DMEM to F12 cell culture media, with additives bFGF [50 ng/ml; Peprotech], EGF [200 ng/ml; Peprotech], B27 supplement [4%; Life Technologies], penicillin/streptomycin [2%; StemCell Technologies] and fungizone [0.4%; Thermo Fisher Scientific]) at a concentration of 100,000 cells/ml. Cells were fed every 3 d with fresh cell culture media and grown for 8–9 d at 37°C in 5% $CO_2$ before passaging. Cells were passaged twice before experimentation. Passaged NSCs were plated in six-well plates in growth media at a density of 10,000 cells/ml and treated with 0.2 µM 4-OHT before analysis at 9 d after the second passage. Neurospheres were counted and imaged to measure their mean diameters in each well.

## Spermatogonial culture

Spermatogonia were isolated from 6- to 8-d-old pup testis by magnetic assisted cell sorting using a Thy1.2 (CD90.2) rat anti-mouse antibody and cultured as previously described (Yeh et al, 2007). To generate *Pnkp*$^{fl/fl}$ deletions in spermatogonia cultures, 0.2 µM 4-OHT was used; with the latter showing no effect on control spermatogonial viability or cell numbers. Spermatogonia were cultured on STO feeder cells as previously described (Yeh et al, 2007) for the first five passages to establish the cultures and then passaged onto 1:10 reduced factor Matrigel-coated wells before treatment in order to reduce the number of feeder cells in the cultures. Cells were treated on day 3 in culture and collected or passaged on day 6. For extended culture experiments, a second 4-OHT treatment was added on day 7 (1 d after passage) and cells were then collected on day 11. Cells were seeded at a density of $2.0 \times 10^5$ cells/ml in 12- or 24-well plates.

## Flow cytometry

Cultured spermatogonia were fixed in 4% PFA and probed with the following primary antibodies and appropriate species-specific secondary antibodies: CASP3 at a 1:800 dilution and γH2AX at a 1:800 dilution (both were rabbit monoclonal antibodies from Cell Signaling Technologies). Rabbit IgG isotype controls were used for gating. Proliferation was measured using EdU incorporation (1 mg/ml) which was added to cell culture medium for 6 h before collection and staining according to the manufacturer's instructions (Click-iT EdU; Invitrogen). Cells were analyzed using a BD FACSCalibur.

## Quantitative PCR

RNA was isolated from pelleted cells using QIAGEN RNeasy Mini Kit followed by reverse transcription using ABI High Capacity cDNA Synthesis kit according to the manufacturer's instructions. cDNA samples were amplified using Bio-Rad SosoFast EvaGreen master mix and gene specific primers as follows: *Pnkp* forward 5′-TTC AGG TGC TGG TGG CAA CA-3′ reverse 5′-CTG CTG CAT CTC CCA CGA AGA C-3′, *Id4* forward 5′-ACT CAC CGC GCT CAA CAC T-3′ reverse 5′-GCA CAC CTG GCC ATC CAT-3′, *Gfrα1* forward 5′-AGA GCC TGC AGG AAA TGA CCT ACT-3′ reverse 5′-GCA TCG AGG CAG TTG TTC CCT T-3′, *Sohlh1*

forward 5′-GGA TGC AGC AAG ACT CCT CAG C-3′ reverse 5′-CTG ACC ACG TTT CTC CGA AGA CTG-3′, *cKit* forward 5′-CCT CAG CCT CAG CAC ATA GC-3′ reverse 5′-CTG GCG TTC ATA ATT GAA GTC ACC-3′ and *Hprt1* forward 5′-AGC AGT ACA GCC CCA AAA TGG T-3′ reverse 5′-CCA ACA AAG TCT GGC CTG TAT CC-3′. Samples were run on an ABI 7500 Fast qPCR and relative fold change was calculated using ΔΔCT relative to housekeeping gene *Hprt1*.

## CellROX flow cytometry methods

Oxidative stress was measures using CellROX (Invitrogen) according to the manufacturer's instructions. Briefly, 5 µM of CellROX Green Reagent was added to spermatogonia cell cultures and incubated at 37°C for 30 min, wells were washed three times with PBS, and fixed in 4% formaldehyde for 15 min. Spermatogonia cells were then analyzed using a BD FACSCalibur and FlowJo software.

## Immunoblotting

Lysates were prepared from control and *PNKP* KO HCT116 cells (Chalasani et al, 2018) in RIPA buffer (Cat. no. 89900; Thermo Fisher Scientific). Then, 30 µg of protein was loaded onto each lane of a 10% resolving gel and then transferred onto nitrocellulose membrane (Cat. no. 1620115; Bio-Rad Laboratories) using the Trans-Blot Turbo transfer system (Bio-Rad Laboratories). The membrane was blocked with 5% milk in 1× TBST buffer for 1 h at room temperature, and then it was cut into three pieces for immunoblotting for PNKP, with b-actin and XPF serving as loading controls. Primary antibodies (anti-PNKP, 1:1,000, dilution, sc-365724; Santa Cruz; anti-β-Actin − 1:3,000, sc-47778; Santa Cruz; anti-XPF, 1:1,000 dilution, MA5-12060; Thermo Fisher Scientific) were prepared in 5% milk in 1× TBST buffer and the membranes were incubated overnight at 4°C. Membranes were then washed 3× in 1× TBST followed by a 1 h incubation with the secondary antibody (HRP goat anti-mouse IgG, dilution 1:3,000, 926-80010; LI-COR) at room temperature. After three washes in 1× TBST, Super Signal West Pico Plus Chemiluminescent Substrate (Cat. no. 34577; Thermo Fisher Scientific) was applied prior to examination using an Odyssey scanner (LI-COR). Images were processed and quantified using Image Studio (Ver. 5.2; LI-COR).

## Crystal violet assay

Wells in a 96-well plate were seeded with 3,000 cells per well and incubated overnight to allow cell attachment. Serial dilutions of a 10 mM (in 100% ethanol) stock solution of 4-OHT (H7904; Sigma-Aldrich) were prepared in cell culture media and added to the plates. A crystal violet based assay was performed after a 72-h exposure to 4-OHT. Following exposure to the drug, the media was discarded, attached cells were stained with 0.5% crystal violet (Sigma-Aldrich) for 20 min at room temperature, washed three times with distilled water and allowed to air-dry. Crystal violet stained cells were shaken for 20 min in 200 µl of methanol and the OD was measured at 570 nm using a FLUOstar OPTIMA microplate reader (BMG Labtech). Percent cell survival was calculated by subtracting the OD of blank wells and then normalizing DMSO controls to 100%. Data were plotted using GraphPad Prism 5.04 (GraphPad Software).

## Colony forming assay

Between 200 and 3,000 control and *PNKP* KO HCT116 cells were seeded in 60-mm petri dishes and incubated overnight to allow attachment. On day 2, cells were exposed to various concentrations of 4-OHT (10 mM stock in ethanol, diluted in culture media). After 9 d incubation, colonies were fixed, stained and counted. To measure their sensitivity to  ionizing radiation, cells were irradiated with $^{60}$Co g-radiation using a Gamma Cell Irradiator (Atomic Energy of Canada Ltd.) using the following doses: 0, 1, 2, 4, 6, and 8 Gy. After 9 d incubation, colonies were fixed, stained and counted. Data were plotted using GraphPad Prism 5.04.

## Statistics

Quantification of immunohistochemistry sections was performed by an evaluator blinded as to the identities of the samples. Quantitative data were represented as means with SD. Independent samples *t* tests and two-way ANOVA were performed according to statistical guidelines in SPSS Statistics 24. Significance was set at $P < 0.05$.

# Data Availability

No data to be disclosed.

# Supplementary Information

# Acknowledgements

We are grateful to Cameron Fielding the University of Calgary Centre for Genome Engineering in the Clara Christie Centre for Mouse Genomics for generating the 'floxed' *Pnkp* transgenic line. We also thank Dragana Ponjevic for her assistance in preparing some of the tissue samples for histological analysis. Funding that enabled purchase of the Zn finger nucleases used in this study was kindly provided by the Cumming School of Medicine Centre for Advanced Technologies. This work was supported by grants from the Canadian Institutes of Health Research (J Biernaskie, PJT159487; FR Jirik, MOP342685; M Weinfeld, PJT168869).

## Author Contributions

W Shin: formal analysis, validation, investigation, methodology, and writing—original draft, review, and editing.
W Alpaugh: formal analysis, validation, investigation, methodology, and writing—original draft.
L Hallihan: resources and methodology.
S Sinha: validation, investigation, and methodology.
E Crowther: validation, investigation, and methodology.
GR Martin: investigation and methodology.
T Scheidl-Yee: investigation and methodology.
X Yang: formal analysis and investigation.
G Yoon: validation, investigation, and methodology.
T Goldsmith: validation.
ND Berger: validation and writing—review and editing.
LGN de Almeida: investigation.
A Dufour: investigation and methodology.
I Dobrinski: conceptualization, supervision, project administration, and writing—review and editing.
M Weinfeld: conceptualization, supervision, funding acquisition, project administration, and writing—review and editing.
FR Jirik: conceptualization, supervision, funding acquisition, project administration, and writing—original draft, review, and editing.
J Biernaskie: conceptualization, supervision, funding acquisition, project administration, and writing—review and editing.

## Conflict of Interest Statement

The authors declare that they have no conflict of interest.

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
