## [Reviewer comments · Life Science Alliance]

Life Science Alliance

PNKP is required for maintaining the integrity of progenitor cell populations in adult mice

Wisoo Shin, Whitney Alpaugh, Laura Hallihan, Sarthak Sinha, Emilie Crowther, Gary R Martin, Teresa Scheidl-Yee, Xiaoyan Yang, Grace Yoon, Taylor Goldsmith, N Daniel Berger, Luiz Gustavo Nogueira de Almeida, Antoine Dufour, Ina Dobrinski, Michael Weinfeld, Frank R Jirik and Jeff Biernaskie

DOI: <https://doi.org/10.26508/lsa.202000790>

Corresponding author(s): Prof. Jeff Biernaskie (University of Calgary) and Dr. Frank R Jirik (University of Calgary)

Review

Timeline:

Submission Date:	2020-05-21
Editorial Decision:	2020-06-25
Revision Received:	2020-11-19
Editorial Decision:	2020-12-23
Appeal Received:	2021-06-04
Editorial Decision:	2021-06-10
Revision Received:	2021-06-15
Editorial Decision:	2021-06-22
Revision Received:	2021-06-23
Accepted:	2021-06-23

Transaction Report:

June 25, 2020

Re: Life Science Alliance manuscript #LSA-2020-00790

Author information redacted

Dear Dr. Biernaskie,

Thank you for submitting your manuscript entitled "PNKP is required for maintaining the integrity of progenitor cell populations in adult mice" to Life Science Alliance. The manuscript was assessed by expert reviewers, whose comments are appended to this letter.

While the reviewers are overall supportive, they have raised concerns regarding the effects of Cre and Tamoxifen alone in a setting where DNA repair is impaired. We understand that addressing this concern is experimentally not trivial. However, we do suggest to i) explicitly discuss this caveat and to ii) use WT and Pnkp KO neurospheres/stem cells/spermatagonia in culture, by adding tamoxifen and/or adding straight Cre and analysing for example transient elevation in poly(ADP-ribose) as a measure of DNA damage. This may represent a feasible experimental approach to probe if DNA damage in the settings used is induced by tamoxifen and/or Cre alone.

We are aware that many laboratories cannot function fully during the current COVID-19/SARS-CoV-2 pandemic and therefore encourage you to take the time necessary to revise the manuscript to the extent requested above. We will extend our 'scooping protection policy' to the full revision period required. If you do see another paper with related content published elsewhere, nonetheless contact us immediately so that we can discuss the best way to proceed.

Thank you for this interesting contribution to Life Science Alliance. We are looking forward to receiving your revised manuscript.

Sincerely,

Reilly Lorenz
Editorial Office Life Science Alliance
Meyerhofstr. 1
69117 Heidelberg, Germany
t +49 6221 8891 414
e contact@life-science-alliance.org
www.life-science-alliance.org

B. MANUSCRIPT ORGANIZATION AND FORMATTING:

Reviewer #1 (Comments to the Authors (Required)):

The manuscript by Shin et al describes a new mouse model in which Pnkp is deleted in the adult mouse by tamoxifen-induced Cre. The authors detect defects in a variety of tissues (brain, skin,

testes) consistent with defects/loss in a variety of progenitor cell types. Overall the paper adds new information to our understanding of the physiological importance of Pnkp, beyond our knowledge gleaned from conditional knockouts and from human diseases associated with PNKP mutation. The findings are intriguing, and consistent with a critical role for Pnkp in progenitor cell maintenance and proliferation, and so perhaps relevant to ageing. Mouse histopathology is not my area of expertise, and so I cannot comment too much on the quality of the pathological analyses. However, the manuscript would be much improved by the addition of some mechanistic/molecular data. In particular there are few if any cultured/isolated cell models in which PNKP is completely deleted, because of the severe impact of complete Pnkp loss on cell proliferation. The authors seem to have a few such cell models here, including neuroblasts/neural stem cells and spermatogonia. It would be useful to examine the level of SSB and DSB repair capacity in such cells. There is currently some lack of clarity in the field in terms of the nature and level of the repair defect in PNKP defective cells, which may relate to the amount of residual PNKP activity (e.g. in human patient cells) and/or tissue/cell type specificity. The authors have a unique opportunity to address these questions greatly enhancing the utility and values of their new mouse model.

Reviewer #2 (Comments to the Authors (Required)):

Evaluation of manuscript entitled: "Polynucleotide kinase phosphatase (PNKP) is required for maintaining the integrity of progenitor cell populations in adult mice" by Shin W. et al,

Synopsis, significance and overall assessment

The manuscript by Shin W. and others concerns the biomedical significance of defects in the DNA repair enzyme PNKP, which are linked with a variety of neurodevelopmental disorders in humans due to autosomal recessive or compound heterozygous mutations. However, the embryonal lethality of full Pnkp inactivation hampers analysis of the role of this important base excision repair (BER) protein in neuronal development and neurodegeneration in mice. Therefore, the authors have generated a conditional floxed Pnkp allele, and crossed it with a tamoxifen-inducible ubiquitous UBC-CRE- transgene, that can cause complete inactivation of the Pnkp gene at any moment after development. They report that Pnkp loss produced a number of segmental premature aging-like phenotype linked with increased genomic instability, including loss of progenitor cells in hair follicles, spermatogonia, and neural progenitors.

The manuscript presents a substantial amount of work on a very relevant and interesting topic: the role of genome stability in the process of aging and stem cell maintenance. However, although the principal findings on the role of Pnkp in various progenitor populations may be valid, there are several potential complications inherent to the experimental design using tamoxifen to activate the UBC-Cre promoter for the deletion of the floxed Pnkp gene and additional comments to the manuscript.

Concerns related to the experimental set-up:

1. For efficient inactivation of Pnkp the tamoxifen should reach all parts of the body in sufficient concentrations. This is the rationale for the 5-fold consecutive tamoxifen injections as done in the experiments presented (Fig. 1F). The immunoblot in Fig. 1G and H nicely shows very low Pnkp protein signals in liver, brain and kidney, which are highly vascularized organs and tissues. However, the analysis is on bulk tissue. It does not exclude the presence of a subfraction of still wt (stem)cells, which might later repopulate specific (parts of organs) organs, masking the phenotype. Moreover, some tissues are relatively poorly vascularized and may be less efficiently reached by the

tamoxifen inducer, such as the epidermis, which may explain the lack of apoptosis upon Pnkp depletion in the interfollicular dermis, as compared to the hair follicle (Fig. 2B).

2. A more critical aspect is that the 5 consecutive tamoxifen injections used to activate the Ubc-Cre promoter for deleting the floxed Pnkp gene, in fact also cause DNA lesions since tamoxifen is a DNA-damaging agent itself, which is not generally realized in mouse transgenesis as the gene-inactivation is usually done in wt (repair-proficient) mice, that may repair most of the DNA lesions. However, this may be of much more importance in the case of repair defects. There is ample evidence (and a lot of literature) that tamoxifen causes significant amounts of DNA damage. This includes a series of tamoxifen (4-OHT)-DNA adducts, at least some of which appear to be repaired by nucleotide excision repair (e.g. see S Shibutani, J T Reardon, N Suzuki, A Sancar, Excision of tamoxifen-DNA Adducts by the Human Nucleotide Excision Repair System, *Cancer Res.* 2000 May 15;60(10):2607-10, and K I E McLuckie, R J R Crookston, M Gaskell, P B Farmer, M N Routledge, E A Martin, K Brown, Mutation Spectra Induced by alpha-acetoxytamoxifen-DNA Adducts in Human DNA Repair Proficient and Deficient (Xeroderma Pigmentosum Complementation Group A) Cells. *Biochemistry* 2005 Jun 7;44(22):8198-205). However, it also leads to DNA breaks which is specifically relevant for BER and particularly for Pnkp deficiency. It has even been reported that tamoxifen treatment of wt mice can cause male infertility (e.g. see Sadeghu S et al., *J. Reprod. Infert.* 20, 10-15, 2019), which by itself is relevant in view of the observation in the current manuscript that absence of spermatogenesis is one of the most prominent features reported here for Pnkp-deficient mice (Fig. 1K, Suppl. Fig 1F,G, Fig. 4).

3. Moreover, it is known that CRE, besides incising the flox sequences, also causes DNA breaks at aspecific sites which are known to cause chromosomal aberrations. In normal mice this may not be extremely important, but may be more deleterious in Pnkp-deficient mice/cells, compared to wt repair-competent animals/cells.

The above concerns are not only relevant for the mouse phenotype reported, but are also consistent with the experimental findings involving spermatogonia progenitors in culture treated with 4-OHT (Fig. 4F-K) and the self-renewal capacity of neural progenitors in vitro cultures (Fig. 5H-L). Hence, from the current results it cannot be excluded that the Pnkp repair deficiency makes cells and the organism both more sensitive to the tamoxifen treatments as well as the CRE activation, which may contribute to the observed phenotype and thereby the main message of the manuscript. One possible experiment that could shed light on whether this scenario is valid would be to carry out a tamoxifen treatment at a later time in an animal that is already Pnkp-deficient and follow whether the phenotype deteriorates by monitoring e.g. body weight, food intake, health status, lifespan etc. compared to untreated Pnkp-KO animals and a wt animal treated with tamoxifen.

In this regard the preferred experimental set-up for the study would have been to include from the beginning besides wt mice two important additional controls, Pnkp^{f/f} without Ubc-CRE transgene, and Wt containing the Ubc-CRE transgene both treated with tamoxifen, to examine whether CRE activation and tamoxifen themselves have any effect on the tissues and organs investigated here. Ideally, one would like to examine Pnkp-KO cells and mice obtained without Tamoxifen pretreatment and without CRE expression and subject these to both separately and in combination.

Additional Comments

The phenotypic description of the conditional Pnkp mouse mutant is quite limited, raising obvious questions in relation to the main conclusion and the connection with stem/progenitor cells and

aging. What is the effect of Pnkp deficiency on life span, what is their cause of death and to which extent is the phenotype progressive as one would expect for accelerated aging?

In relation to the stem progenitor cell phenotype an obvious question is what happens to the intestinal stem cells which belong to the most proliferative cells of the entire soma and to the hematopoietic system? Is proliferation in general reduced?

Is besides apoptosis also senescence induced?

Minor comments and questions

Suppl. Fig. 1: indicate the age of the mice shown. Are body weight curves available to demonstrate the extent of reduced weight gain or of weight loss and whether this phenotype is progressive or remains the same throughout life. Also, it seems that besides the strong reduction of the subcutaneous fat layer in the PnkpKO mice the dermis is much thicker (compare Suppl. Fig. 1 panels B and C). Were the four mutant stillborn pups of the Nestlin-Cre driven Pnkp inactivation in any regard abnormal?

At several places mice or tissues of mice are shown without indication of age, which for accelerated aging mutants is a highly relevant piece of information (e.g. Fig. 3, several panels, Suppl. Fig. 1, age only indicated for panels F and G).

The figure order in the text and in the figures should preferably be congruent. E.g. in the manuscript the order of discussing Fig. 2 is Fig. 2A, then Fig. 2G followed by 2F, then B and D and finally C and E. etc. Also, for Fig. 1, Fig 1M is only discussed after Fig 3C and Fig. 1L is not discussed at all in the text.

The paragraph in the Results entitled: "Increased DNA damage, ROS levels, and apoptosis in spermatogonial progenitors", does not contain any data on ROS levels.

Reviewer #3 (Comments to the Authors (Required)):

This is a relatively straightforward analysis of the consequence of loss of PNKP in proliferative zones in adult mice. The approach has been to drive PNKP deletion with a tamoxifen-inducible UBC-cre. The clear message is that loss of this repair factor compromises viability of proliferating cells. The authors have presented a series of examples involving dermal progenitors, neural progenitors and a main focus is on proliferation/spermatogenesis in the testis. The analysis is descriptive, utilizing standard methodologies to survey histology in the respective tissues under study. Although the results are expected, this is nonetheless a solid study that supports other studies showing how essential PNKP is in replicating cells

Reviewer #1 (Comments to the Authors (Required)):

The manuscript by Shin et al describes a new mouse model in which Pnkp is deleted in the adult mouse by tamoxifen-induced Cre. The authors detect defects in a variety of tissues (brain, skin, testes) consistent with defects/loss in a variety of progenitor cell types. Overall the paper adds new information to our understanding of the physiological importance of Pnkp, beyond our knowledge gleaned from conditional knockouts and from human diseases associated with PNKP mutation. The findings are intriguing, and consistent with a critical role for Pnkp in progenitor cell maintenance and proliferation, and so perhaps relevant to ageing. Mouse histopathology is not my area of expertise, and so I cannot comment too much on the quality of the pathological analyses. However, the manuscript would be much improved by the addition of some mechanistic/molecular data. In particular there are few if any cultured/isolated cell models in which PNKP is completely deleted, because of the severe impact of complete Pnkp loss on cell proliferation.

Mice in which PNKP is deleted when they are young are still capable of growing in size and maintaining themselves for more than a year, thus the lack of PNKP does not completely abrogate all progenitors equally. If the dramatic spermatogonial progenitor depletion due to PNKP loss in the testes was a feature of other progenitor populations (e.g. gut lining, keratinocytes, immune system, and bone marrow), one would have expected that global loss of PNKP would be rapidly lethal. Thus far, we have not directly assessed these cell populations with respect to their function in Pnkp deficient animals.

It could be that different progenitor populations vary in terms of their sensitivity to PNKP loss, for example, differing with respect to their levels of ROS, anti-oxidant defenses, or the expression levels of a backup 3'-phosphatase that has yet to be identified (Chalasani et al. DNA Repair 2018 68:12-24).

Another possibility is that there might be small numbers of stem cells that escape Cre-mediated Pnkp gene deletion (during the short period of early 4-OHT exposure), such that in a tissue like bone marrow that is capable of massive increases in proliferation, any residual PNKP expressing stem cells have a growth advantage and are able to fully reconstitute the tissue.

Regarding the note about cultured cells, we have established tumour cell lines in which PNKP has been completely deleted (via CRISPR-Cas9 mutagenesis, Chalasani et al. DNA Repair 2018 68:12-24) showing that cell growth still occurs and that the lines are also able to generate tumours in vivo when xenografted in mice.

Indeed, for these set of experiments, the *Pnkp* knockdown efficiency was approximately 80% in both spermatogonial (Figure 4K) and neural progenitors (Figure 5L).

The authors seem to have a few such cell models here, including neuroblasts/neural stem cells and spermatogonia. It would be useful to examine the level of SSB and DSB repair capacity in such cells.

It would indeed be very interesting to examine but we feel that is beyond the scope of the current study, and impossible to carry out with University research activities been curtailed.

There is currently some lack of clarity in the field in terms of the nature and level of the repair defect in PNKP defective cells, which may relate to the amount of residual PNKP activity (e.g. in human patient cells) and/or tissue/cell type specificity. The authors have a unique opportunity to address these questions greatly enhancing the utility and values of their new mouse model.

We agree and are hoping to be able to exploit the model further in future experiments.

Reviewer #2 (Comments to the Authors (Required)):

Evaluation of manuscript entitled: "Polynucleotide kinase phosphatase (PNKP) is required for maintaining the integrity of progenitor cell populations in adult mice" by Shin W. et al,

Synopsis, significance and overall assessment

The manuscript by Shin W. and others concerns the biomedical significance of defects in the DNA repair enzyme PNKP, which are linked with a variety of neurodevelopmental disorders in humans due to autosomal recessive or compound heterozygous mutations. However, the embryonal lethality of full Pnkp inactivation hampers analysis of the role of this important base excision repair (BER) protein in neuronal development and neurodegeneration in mice. Therefore, the authors have generated a conditional floxed Pnkp allele, and crossed it with a tamoxifen-inducible ubiquitous UBC-CRE- transgene, that can cause complete inactivation of the Pnkp gene at any moment after development. They report that Pnkp loss produced a number of segmental premature aging-like phenotype linked with increased genomic instability, including loss of progenitor cells in hair follicles, spermatogonia, and neural progenitors.

The manuscript presents a substantial amount of work on a very relevant and interesting topic: the role of genome stability in the process of aging and stem cell maintenance. However, although the principal findings on the role of Pnkp in various progenitor populations may be valid, there are several potential complications inherent to the experimental design using tamoxifen to activate the UBC-Cre promoter for the deletion of the floxed Pnkp gene and additional comments to the manuscript.

Owing to University research having been, in effect, shut-down since the early months of the COVID-19 pandemic, we nevertheless attempted to carry out the approach that you suggested. Due to limited access of staff and hence our ability to carry out *in vitro* wet-lab work and the fact that our mouse colony needed to be downsized. However, we did have a limited residual number of mice of the appropriate genotypes that we were able to acutely treat with 4-OHT and immediately afterwards harvest tissues for subsequent lysate immunoblotting for PAR. More recently as limited access was granted, some immunoblotting experiments were performed. Because we were looking for the general genotoxic effects of 4-OHT, and UBC-CreERT2 plus 4-OHT, we selected a tissue having a somewhat homogenous cellular composition, namely the liver.

As can be seen in the blotting experiments shown below, we found modest elevations in high molecular weight PAR signals in the livers of 4-OHT treated mice that that were also positive for UBC-CreERT2 and floxed Pnkp alleles. Neither UBC-CreERT2 plus 4-OHT nor 4-OHT alone appear to augment PAR signals over controls. A limitation of this approach is that we were only able to study the effects of Pnkp deletion in the presence of both 4-OHT and the activated Cre. Thus, it was not possible to distinguish a PAR signal due to PNKP loss alone, from a PAR signal due to PNKP loss while in the presence of Cre and 4-OHT. It is plausible that PAR backgrounds would be elevated in Pnkp deficient tissues, even in the absence of 4-OHT, since loss of PNKP would sensitize cells to physiological levels of reactive oxygen species.

An important caveat is that DNA injury due to either adventitious (i.e. non-*loxP* site-targeted) Cre activity, or that resulting from 4-OHT induced DNA adduct repair (by the nucleotide excision repair pathway) would not yield substrates requiring PNKP activity. That is because in neither instance will the normal DNA repair mechanisms involved lead to generation of either 3'-PO₄ or 5'-OH termini (i.e. PNKP substrates). A potential exception to this might be DNA damage due to 4-OHT-induced

oxidative stress, as was reported for MCF-7 cells *in vitro* [Wozniak K. et al Arch Toxicol. 2007 Jul;81(7):519-27].

It should be noted that 4-OHT treatment is limited to 5 consecutive days at a time when mice are very young; and yet the phenotype we observe is still evident many months later, and even up to a year. It thus seems extremely unlikely that transient oxidative stress evoked by 4-OHT would be capable of producing a permanent defect in future skin, testes, and brain progenitor populations.

We therefore propose that the immunoblotting results below not be included as a supplemental figure in the manuscript for the reasons that were articulated. It would also potentially give a reader the false impression that the exposure to 4-OHT was responsible for the permanent injury to progenitors.

[Figure removed by editorial staff per authors' request].

Determination of PAR levels by immunoblotting. (A) Representative immunoblot of poly-ADP-ribose (PAR) levels in liver samples from UBC-CreER^{T2}:*Pnkp*^{fl/fl}, WT:WT, *Pnkp*^{fl/fl}:WT, and UBC-CreER^{T2}:WT mice. Samples were collected 72 hours after treatment with 4-hydroxytamoxifen (TMX) in sunflower oil (SO), or SO alone for 5 consecutive days. Lanes were loaded with 30 µg of protein from each tissue and, after transfer, membranes were immunoblotted with the anti-PAR (clone 10H) mouse monoclonal antibody. Loading control was performed by measuring glyceraldehyde-3-phosphate dehydrogenase (GAPDH) levels, assessed on the same blot with the anti-GAPDH rabbit monoclonal antibody. (B) Quantification of total PAR levels by densitometry. PAR bands were visualized via ECL chemiluminescence and then the signal density was quantified using Image J (1.52K National Institute of Health, USA). Semi-quantitative analyses were performed by measuring the relative abundance of the PAR normalized to GAPDH. An identical densitometric measurement area was used for all lanes and the background was subtracted. One-way ANOVA followed by Tukey's multiple comparisons post-hoc test between all groups was used (Prism 8.0). Values are means ± SEM, n = 2 mice per group. In summary: while there were no statistically significant differences between the mice groups or treatment regimens (e.g. TMX vs SO), there may have been a trend for the CreER^{T2}:*Pnkp*^{fl/fl} treated with TMX to express higher levels of the PAR.

Concerns related to the experimental set-up:

1. For efficient inactivation of Pnkp the tamoxifen should reach all parts of the body in sufficient concentrations. This is the rationale for the 5-fold consecutive tamoxifen injections as done in the experiments presented (Fig. 1F). The immunoblot in Fig. 1G and H nicely shows very low Pnkp protein signals in liver, brain and kidney, which are highly vascularized organs and tissues. However, the analysis is on bulk tissue. It does not exclude the presence of a subfraction of still wt (stem)cells, which might later repopulate specific (parts of organs) organs, masking the phenotype.

Mice in which PNKP is deleted when they are young are still capable of growing in size and maintaining themselves for more than a year, thus the lack of PNKP does not completely abrogate all progenitors equally. If the dramatic spermatogonial progenitor depletion due to PNKP loss in the testes was a feature of other progenitor populations (e.g. gut lining, keratinocytes, immune system, and bone marrow), one would have expected that global loss of PNKP would be rapidly lethal. Thus far, we have not directly assessed these cell populations with respect to their function in Pnkp deficient animals.

It could be that different progenitor populations vary in terms of their sensitivity to PNKP loss, for example, differing with respect to their levels of ROS, anti-oxidant defenses, or the expression levels of a backup 3'-phosphatase that has yet to be identified (Chalasan et al. DNA Repair 2018 68:12-24).

Another possibility is that there might be small numbers of stem cells that escape Cre-mediated Pnkp gene deletion (during the short period of early 4-OHT exposure), such that in a tissue like bone marrow that is capable of massive increases in proliferation, any residual PNKP expressing stem cells have a growth advantage and are able to fully reconstitute the tissue.

Moreover, some tissues are relatively poorly vascularized and may be less efficiently reached by the tamoxifen inducer, such as the epidermis, which may explain the lack of apoptosis upon Pnkp depletion in the interfollicular dermis, as compared to the hair follicle (Fig. 2B).

The vascular supply within each of these skin compartments (dermis, epidermis and hair follicle) is entirely sufficient to enable effective tamoxifen induction and recombination of transgenic reporters or excision of floxed alleles. We have used a variety of CreERT2 systems previously in both hair follicles and dermis with excellent success and recombination efficiency, even when titrated down to enable in vivo clonal analysis experiments (Rahmani et al Developmental Cell, 2014; Shin et al Developmental Cell, 2020; Abbasi et al Cell Stem Cell, 2020).

2. A more critical aspect is that the 5 consecutive tamoxifen injections used to activate the UBC-Cre promoter for deleting the floxed PnpK gene, in fact also cause DNA lesions since tamoxifen is a DNA-damaging agent itself, which is not generally realized in mouse transgenesis as the gene-inactivation is usually done in wt (repair-proficient) mice, that may repair most of the DNA lesions. However, this may be of much more importance in the case of repair defects. There is ample evidence (and a lot of literature) that tamoxifen causes significant amounts of DNA damage.

This includes a series of tamoxifen (4-OHT)-DNA adducts, at least some of which appear to be repaired by nucleotide excision repair (e.g. see S Shibusaki, J T Reardon, N Suzuki, A Sancar, Excision of tamoxifen-DNA Adducts by the Human Nucleotide Excision Repair System, Cancer Res. 2000 May 15;60(10):2607-10, and K I E McLuckie, R J R Crookston, M Gaskell, P B Farmer,

M N Routledge, E A Martin, K Brown, Mutation Spectra Induced by alpha-acetoxytamoxifen-DNA Adducts in Human DNA Repair Proficient and Deficient (Xeroderma Pigmentosum Complementation Group A) Cells. Biochemistry 2005 Jun 7;44(22):8198-205). However, it also leads to DNA breaks which is specifically relevant for BER and particularly for Pnkp deficiency. It has even been reported that tamoxifen treatment of wt mice can cause male infertility (e.g. see Sadeghu S et al., J. Reprod. Infert. 20, 10-15, 2019), which by itself is relevant in view of the observation in the current manuscript that absence of spermatogenesis is one of the most prominent features reported here for Pnkp-deficient mice (Fig. 1K, Suppl. Fig 1F,G, Fig. 4).

We observed no abnormality in spermatogonial cell populations when 4-OHT was administered to various control mice. Thus, the dramatic and rapid depletion of testes progenitors was only seen when Pnkp was deleted. The point is well taken, however, that there might be a superimposed effect of tamoxifen that contributes to the PNKP deficient phenotype. See discussion below regarding the theoretical possibility that tamoxifen may induce oxidative stress, and effect that might be predicted to aggravate the consequences of PNKP deficiency.

With respect to tamoxifen and DNA damage induction: It is very unlikely that tamoxifen-DNA base adducts play a role in generating the observed Pnkp(-/-) phenotype, this is because these lesions are repaired by the nucleotide excision repair pathway which does not require PNKP; specifically because the DNA incisions by XPG and XPF result 3'-OH and 5'-phosphate termini which are not PNKP substrates.

On the other hand, there was a paper that reported the generation of single-strand breaks by tamoxifen in MCF7 breast cancer cells (Wozniak K. et al Arch Toxicol. 2007 Jul;81(7):519-27). The dose range in their experiments was 1-10 microM for 1 hr. How might this compare to the dose given to the mice is not clear. Thus, the potential for ROS-induced single-strand breaks caused by tamoxifen might plausibly be an issue because these can include strand break termini that require PNKP for their repair. The reported breaks do appear to be "oxidative" in nature, so the reviewer has raised a good point.

However, as mentioned in the cover letter to the Editor, it should be noted that 4-OHT treatment is limited to 5 consecutive days at a time when mice are very young; and yet the phenotype we observe is still evident many months later, and even up to a year or more. It thus would seem very unlikely that transient oxidative stress exposure evoked by 4-OHT administration, at a time when Pnkp deficient mice are still very young, would be capable of producing a permanent defect that is reflected in future skin, testes, and brain progenitor populations.

3. Moreover, it is known that CRE, besides incising the flox sequences, also causes DNA breaks at specific sites which are known to cause chromosomal aberrations. In normal mice this may not be extremely important, but may be more deleterious in Pnkp-deficient mice/cells, compared to wt repair-competent animals/cells.

This possibility seems unlikely, given that Cre recombinase creates temporary ssDNA breaks with 3'-OH termini such that any un-rejoined breaks arising from Cre would not require PNKP for their repair.

4. The above concerns are not only relevant for the mouse phenotype reported, but are also consistent with the experimental findings involving spermatogonia progenitors in culture treated with 4-OHT (Fig. 4F-K) and the self-renewal capacity of neural progenitors in vitro cultures (Fig. 5H-L). Hence, from the current results it cannot be excluded that the Pnkp repair

deficiency makes cells and the organism both more sensitive to the tamoxifen treatments as well as the CRE activation, which may contribute to the observed phenotype and thereby the main message of the manuscript.

In view of the report that tamoxifen can induce oxidative stress in cultures MCF-7 cells (Wozniak K. et al Arch Toxicol. 2007 Jul;81(7):519-27) this is a good theoretical point since the acute exposure of progenitor cultures to 4-OHT would not only lead to Pnkp gene deletion, but at the same time would potentially lead to some level of oxidative stress that could generate lesions requiring PNKP-mediated repair. Determining whether or not this is in fact occurring would be an interesting possibility for further exploration.

One possible experiment that could shed light on whether this scenario is valid would be to carry out a tamoxifen treatment at a later time in an animal that is already Pnkp-deficient and follow whether the phenotype deteriorates by monitoring e.g. body weight, food intake, health status, lifespan etc. compared to untreated Pnkp-KO animals and a wt animal treated with tamoxifen.

This is a good idea, namely whether adult mice having had Pnkp deleted generally would show a dramatic effect in response to a re-application of TMX. However, in view of the comments above about how PNKP loss would not be expected to exacerbate the DNA damaging effects of effects tamoxifen and Cre this experiment would likely not change the phenotype we have already observed. If tamoxifen in the administration regimen we have used, does indeed induce significant oxidative stress, then a change in phenotype might be observed. We have in fact considered exposing the adult Pnkp-deficient mice to mice to a bona fide ROS generating source such as paraquat since they should be very sensitive to such agents. Such studies would be excellent subjects for future experimentation.

In this regard the preferred experimental set-up for the study would have been to include from the beginning besides wt mice two important additional controls, Pnkp^{fl/fl} without Ubc-CRE transgene, and Wt containing the Ubc-CRE transgene both treated with tamoxifen, to examine whether CRE activation and tamoxifen themselves have any effect on the tissues and organs investigated here. Ideally, one would like to examine Pnkp-KO cells and mice obtained without Tamoxifen pretreatment and without CRE expression and subject these to both separately and in combination.

None of these controls led to any aspect of the phenotype. Both controls (WT with Ubc-Cre gene, and Pnkp^{fl/fl} without Ubc-Cre gene) were used as controls in our experiments.

Additional Comments

The phenotypic description of the conditional Pnkp mouse mutant is quite limited, raising obvious questions in relation to the main conclusion and the connection with stem/progenitor cells and aging. What is the effect of Pnkp deficiency on life span, what is their cause of death and to which extent is the phenotype progressive as one would expect for accelerated aging?

An aging study has not been carried out, while the mice show some features plausibly consistent with aging, we did not see any premature deaths in groups of mice housed for up to 12 mos of age. Setting up an experiment whereby a large cohort of mice could be aged out to the time of spontaneous death, with further investigations as to the cause(s) of death (e.g. malignancies, neurodegeneration), would obviously be very interesting to carry out.

In relation to the stem progenitor cell phenotype an obvious question is what happens to the intestinal stem cells which belong to the most proliferative cells of the entire soma and to the hematopoietic system? Is proliferation in general reduced?

This important point was addressed in our response to Reviewer 1.

Is besides apoptosis also senescence induced?

This is a good question, however, we have not addressed this possibility experimentally thus far.

Minor comments and questions

Suppl. Fig. 1: indicate the age of the mice shown. Are body weight curves available to demonstrate the extent of reduced weight gain or of weight loss and whether this phenotype is progressive or remains the same throughout life. Also, it seems that besides the strong reduction of the subcutaneous fat layer in the PnkpKO mice the dermis is much thicker (compare Suppl. Fig. 1 panels B and C).

Thank you for pointing out the potential increase in dermal thickness of KO animals in Figure S1C,D (previously S1B,C). However, this pattern is unique to the image presented and is not representative across mice. Although there are clear differences in the adipose composition in the knockout mice, examination of the upper dermal layer across mice did not show reliable changes in thickness as a consequence of genotype. A systematic analysis that incorporates age, sex matched mice and synchronized hair cycle stage would be required to evaluate this definitively, which we have not done here (but hope to do in future work).

The age of the mice (8 mo) has been added to the Fig S1 caption. Unfortunately, we do not have sufficient data to produce a body weight curve. However, we have added a bar graph comparing the weight of WT and KO animals at 8 mos of age to **Figure S1B**. We only have data for 8-mo old mice as weight was only taken during the initial assessment of mice. The Nestin-Cre animals were badly decomposed when discovered in the cages, and hence a necropsy was not possible. Studies by others (the McKinnon group) have shown that Nestin-Cre excision of Pnkp leads to a severe CNS developmental defect.

At several places mice or tissues of mice are shown without indication of age, which for accelerated aging mutants is a highly relevant piece of information (e.g. Fig. 3, several panels, Suppl. Fig. 1, age only indicated for panels F and G).

Ages of the mice have now been added to Fig 3 and Fig S1. All figure legends have been reviewed so as to include ages of mice in the images.

The figure order in the text and in the figures should preferably be congruent. E.g. in the manuscript the order of discussing Fig. 2 is Fig. 2A, then Fig. 2G followed by 2F, then B and D and finally C and E. etc. Also, for Fig. 1, Fig 1M is only discussed after Fig 3C and Fig. 1L is not discussed at all in the text.

Figure order now matches the order they are referenced in the manuscript.

The paragraph in the Results entitled: "Increased DNA damage, ROS levels, and apoptosis in spermatogonial progenitors", does not contain any data on ROS levels.

The mention of 'ROS levels' in this sentence has now been removed.

Reviewer #3 (Comments to the Authors (Required)):

This is a relatively straightforward analysis of the consequence of loss of PNKP in proliferative zones in adult mice. The approach has been to drive PNKP deletion with a tamoxifen-inducible UBC-cre. The clear message is that loss of this repair factor compromises viability of proliferating cells. The authors have presented a series of examples involving dermal progenitors, neural progenitors and a main focus is on proliferation/spermatogenesis in the testis. The analysis is descriptive, utilizing standard methodologies to survey histology in the respective tissues under study. Although the results are expected, this is nonetheless a solid study that supports other studies showing how essential PNKP is in replicating cells

We thank the reviewer for their comments.

December 23, 2020

Re: Life Science Alliance manuscript #LSA-2020-00790R

Author information redacted

Dear Dr. Biernaskie,

Thank you for submitting your manuscript entitled "PNKP is required for maintaining the integrity of progenitor cell populations in adult mice". The manuscript has been evaluated by expert reviewers, whose reports are appended below. Unfortunately, after an assessment of the reviewer feedback, our editorial decision is against publication of the manuscript in its current form in Life Science Alliance (LSA).

Although your manuscript remains intriguing, the reviewers were not satisfied with the revisions made. While we are sympathetic about the current COVID-related lab situation, the concerns about Cre and Tamoxifen effects raised by the reviewers in both rounds of review, and laid out in our previous decision letter are valid for this study, and will have to be addressed.

Typically, it is our policy to allow only one revision at LSA. However, given your special circumstance, we can make an exception; we are open to resubmission to Life Science Alliance of a significantly revised and extended manuscript that addresses the points laid out in our previous decision letter. The revision will be subject to further peer-review. If you would like to resubmit this work to Life Science Alliance, please submit an appeal directly through our manuscript submission system when the revision is ready. Please note that priority and novelty would be reassessed at resubmission.

Regardless of how you choose to proceed, we hope that the comments below will prove constructive as your work progresses. We would be happy to discuss the reviewer comments further once you've had a chance to consider the points raised in this letter.

Thank you for thinking of Life Science Alliance as an appropriate place to publish your work.

Sincerely,

Shachi Bhatt, Ph.D.
Executive Editor
Life Science Alliance
<https://www.lsjournal.org/>
Tweet @SciBhatt @LSAJournal

Reviewer #1 (Comments to the Authors (Required)):

Disappointingly, the authors have not conducted any of the suggested revisions in respect to increasing the mechanistic insight, which it completely lacks. Whilst it therefore remains entirely descriptive in nature, the work does fall into the remit of LSA, however, and I do appreciate the difficulties imposed by COVID-related shutdown. At the very least, if published, the manuscript

should certainly discuss in detail the limitations of their experiments, in terms of partial PNKP knockout/heterogeneity intrinsic to their mouse model, and the caveat that tamoxifen-induced DNA damage might contribute to their phenotype.

Reviewer #2 (Comments to the Authors (Required)):

It is a real pity and unfortunate that most of the requested experiments apparently could not be conducted due to Covid19 restrictions. This applies to the points raised by ref. 1 the experiments using in vitro cultured cell models (including neuroblasts/neural stem cells and spermatogonia) for assessing repair parameters, which would have provided clarification of the nature and level of the repair defect in PNKP-defective cells, which may relate to the amount of residual PNKP activity. In fact, this system would also have been very informative for investigating the main concerns of this referee (ref. 2) namely that the tamoxifen treatment causes DNA damage that in combination with the PNKP deficiency is responsible for (part of) the phenotype of the mouse mutants, affecting the main message of the manuscript.

Concerning one of my main concerns: the induction of DNA damage by the tamoxifen (4-OHT) treatment the authors were able to carry out some preliminary tests, but appeared unable to perform conclusive experiments, as also recognised by the authors. The finding shown in the figure in the rebuttal, in which High MW PAR was clearly (and as I estimate from the blot significantly) elevated only in the case of 4-OHT treated mice that were also positive for UBC-CreERT2 and floxed Pnkp alleles is entirely consistent with the induction of DNA damage by the 4-OHT that is not or poorly repaired when PNKP is absent (see figure Panel A in rebuttal, first experimental lane compared to all others). However, due to the absence of critical controls this interpretation is speculative. The fact, that the consequences of the 5 consecutive 4-OHT treatments may appear several months later might be explained by e.g., effects on stem cell pools becoming earlier exhausted, which will become manifest only after considerable time. In this regard I disagree with the authors. Also, the cellular studies suggested: "Ideally, one would like to examine Pnkp-KO cells and mice obtained without Tamoxifen pretreatment and without CRE expression and subject these to both separately and in combination." are lacking and the response by the authors to this suggestion is inadequate, because no PNKP-KO cells or mice without tamoxifen pretreatment have been studied. Also, this alternative interpretation of the findings is not at all mentioned in the manuscript.

Although I am very sorry for the authors, it is an unfortunate severe omission that the Tamoxifen option is not mentioned in the manuscript and is not ruled out. In fact, this referee considers the available data consistent and certainly not inconsistent with this possibility.

Reviewer #3 (Comments to the Authors (Required)):

I don't have any further issues or concerns.

Dear Dr. Bhatt,

We wish to thank you and the reviewers for their thoughtful comments and your understanding of the current practical issues arising from the Covid-19 pandemic.

As per my previous earlier correspondence regarding our manuscript (March 9, 2021), we have now completed revision experiments that we think will address the reviewer concerns outlined previously. We would like to apologize to the reviewers, we in no way meant to disregard their concerns in our previous revision.

Below, we have addressed each of the suggestions raised by reviewers below and we feel the manuscript is substantially improved. We have added a caveat section to outline some of the limitations of our study. Below, we also clarify key methodological differences in the examples provided by the reviewer (regarding their suggestion that tamoxifen negatively impacts spermatogenesis or induces DNA adducts). We believe that we have provided a strong case against any confounding contribution of tamoxifen to the observed phenotypes.

Thank you again for your consideration and we look forward to hearing from you.

Sincerely,
Jeff Biernaskie, PhD

June 10, 2021

MS: LSA-2020-00790R

Dear Dr. Biernaskie,

Your manuscript entitled "PNKP is required for maintaining the integrity of progenitor cell populations in adult mice" has now been reconsidered, and I am pleased to let you know that we have accepted your appeal.

We will let you know when a decision has been made.

Please use the following link to submit your manuscript:

<https://lsa.msubmit.net/cgi-bin/main.plex?el=A2Na4Uj4B6CKuU5I3B9ftdfdGnVPshRJb3q18rAgasAZ>

Yours sincerely,
Eric Sawey, PhD
Executive Editor
Life Science Alliance
<http://www.lsjournal.org>

Author responses are highlighted, and corresponding sections in the manuscript are also highlighted

Reviewer #1 (Comments to the Authors (Required)):

Disappointingly, the authors have not conducted any of the suggested revisions in respect to increasing the mechanistic insight, which it completely lacks. Whilst it therefore remains entirely descriptive in nature, the work does fall into the remit of LSA, however, and I do appreciate the difficulties imposed by COVID-related shutdown. At the very least, if published, the manuscript should certainly discuss in detail the limitations of their experiments, in terms of partial PNKP knockout/heterogeneity intrinsic to their mouse model, and the caveat that tamoxifen-induced DNA damage might contribute to their phenotype.

We thank the reviewer for recognizing our intent to limit the scope to a descriptive manuscript. The significance of our research is highlighted in: 1) the generation and characterization of an inducible global Pnkp knockout mouse model that enables interrogation of PNKP function in postnatal organ function and 2) using this model, we show that Pnkp deficiency results in perturbation of multiple progenitor cell populations (but not all) across different organs.

We apologize to the reviewer for not directly addressing the main concerns of the original review. Here, we present three additional experiments to address the concerns raised by the reviewer, specifically addressing the potential confounding effect of tamoxifen-induced DNA damage.

1. As the reviewer suggests, either tamoxifen treatment or loss of *Pnkp* function could independently induce DNA damage and that the combination may have an additive effect that contributes to the observed phenotype (defective progenitor function and eventual loss). However, our revision experiments suggest that tamoxifen is having minimal (if any) impact on the observed phenotype. To test this directly, we treated cancer cell lines after *Pnkp* deletion with increasing doses of tamoxifen and assessed their sensitivity in comparison to wt cancer cells by assaying cell viability, and proliferation potential based on clonogenic activity (Fig S3, lines 381-393). We found that the loss of *Pnkp* does not increase sensitivity of the cells to tamoxifen, even at high concentrations (Fig S3A,B). The dosages used for this experiment were up to 100-fold higher (20 μ M) than those used in our cell culture experiments (0.2 μ M). On the other hand, the loss of *Pnkp* led to increased sensitivity to ionizing radiation (Fig S3C). Therefore, our data suggests the additive effect of tamoxifen and *Pnkp* KO is minimal, and likely impact separate pathways in DNA damage and repair.
2. In order to address the potential of Cre and tamoxifen to induce the progenitor and tissue-level phenotypes observed *in vivo*, we examined a new line of mice (α SMACre:ROSA^{YFP}:Pnkp^{f/f}) (Fig S2A, lines 229-244). We have published extensively using this transgenic model to genetically target a population of dermal progenitors that reside in hair follicles and are essential for maintaining the inductive mesenchyme that supports continuous hair growth/regeneration (Rahmani et al Developmental Cell, 2014; Shin et al Developmental Cell, 2020; Abbasi et al 2020 Cell Stem Cell). These Cre⁺ Pnkp^{f/f} mice were also treated with high doses of tamoxifen (1 mg/day/mouse, once a day for 5 days), yet did not develop any form of hair loss or hyperpigmentation for up to 12 mos (Fig S2B-E). The total amount of tamoxifen was identical to the amount administered to our UBCCre:Pnkp^{f/f} mice. Additionally, although there is robust TAM-induced Cre-mediated recombination that results in both deletion of PNKP

and expression of the YFP reporter, these cells persist within their niche for at least 2 months (Fig S2F). This demonstrates two things: first, combining PNKP deletion and tamoxifen/Cre activation in the same cell does not lead to irreversible DNA damage and cell death. Second, it suggests the crucial function of PNKP may be specific to certain types of tissues or stem/progenitor pools.

3. Lastly, the treatment of spermatogonia with 4-OHT did not lead to an increase in oxidative stress response (Figure 4L, lines 338-340). This data further demonstrates that tamoxifen concentrations used in our *in vitro* experiments do not exacerbate DNA damage or cause excess detriment to cell health.

We believe that this new data sufficiently alleviates the assertion that tamoxifen treatment may contribute to the observed progenitor dysfunction, and tissue degeneration phenotypes we have observed following PNKP gene deletion.

Lastly, we agree with the reviewer regarding the importance of discussing the limitations of our study. As such, we have clarified the potential pitfalls in the discussion (lines 434-451 and lines 289-290). However, the papers cited in the reviewer's comments raise more questions. In the cited paper, the mice were treated continuously with tamoxifen in drinking water for 35 days to drive an effect on spermatogenesis (Sadeghi et al., 2019). Our mice, on the other hand, were only treated with tamoxifen for 5 days. The studies cannot be compared due to the drastic differences in protocol. Indeed, it is not surprising that detrimental effects were observed following chronic administration of tamoxifen. Additionally, the studies reporting that tamoxifen induces DNA adducts were performed by applying tamoxifen directly on to frozen cell extracts (Shibutani et al., 2000) or directly on plasmids (McLuckie et al., 2005). Again, these studies are not comparable to systemic low dose treatment of mice and living cells as we have done in our experiments. **We hope that the reviewer can recognize the major differences in experimental methodology between our study and the studies that have been referenced in their review and consider them accordingly.**

Reviewer #2 (Comments to the Authors (Required)):

It is a real pity and unfortunate that most of the requested experiments apparently could not be conducted due to Covid19 restrictions. This applies to the points raised by ref. 1 the experiments using in vitro cultured cell models (including neuroblasts/neural stem cells and spermatogonia) for assessing repair parameters, which would have provided clarification of the nature and level of the repair defect in PNKP-defective cells, which may relate to the amount of residual PNKP activity. In fact, this system would also have been very informative for investigating the main concerns of this referee (ref. 2) namely that the tamoxifen treatment causes DNA damage that in combination with the PNKP deficiency is responsible for (part of) the phenotype of the mouse mutants, affecting the main message of the manuscript.

Please see above for our response to Review 1.

Concerning one of my main concerns: the induction of DNA damage by the tamoxifen (4-OHT) treatment the authors were able to carry out some preliminary tests, but appeared unable to perform conclusive experiments, as also recognised by the authors. The finding shown in the figure in the rebuttal, in which High MW PAR was clearly (and as I estimate from the blot significantly) elevated only in the case of 4-OHT treated mice that were also positive for UBC-CreERT2 and floxed Pnkp alleles is entirely consistent with the induction of DNA damage by the 4-OHT that is not or poorly repaired when PNKP is absent (see figure Panel A in rebuttal, first experimental lane compared to all others). However, due to the absence of critical controls this interpretation is speculative. The fact, that the consequences of the 5 consecutive 4-OHT treatments may appear several months later might be explained by e.g., effects on stem cell pools becoming earlier exhausted, which will become manifest only after considerable time. In this regard I disagree with the authors. Also, the cellular studies suggested: "Ideally, one would like to examine Pnkp-KO cells and mice obtained without Tamoxifen pretreatment and without CRE expression and subject these to both separately and in combination." are lacking and the response by the authors to this suggestion is inadequate, because no PNKP-KO cells or mice without tamoxifen pretreatment have been studied. Also, this alternative interpretation of the findings is not at all mentioned in the manuscript. Although I am very sorry for the authors, it is an unfortunate severe omission that the Tamoxifen option is not mentioned in the manuscript and is not ruled out. In fact, this referee considers the available data consistent and certainly not inconsistent with this possibility.

We agree that the PAR study was suboptimal and even premature to have included (this was done at the suggestion of one of the previous LSA Editors that reviewed the penultimate submission), and although potentially showing a trend for increased damage on the Pnkp KO, we feel that this was inconclusive, not only given the minimal number of biological replicates, but also the lack of other corroborative evidence of DNA damage being higher in the Pnkp KO. We feel that the a PAR study would require much in the way of additional, and more thorough investigation, perhaps including more than one tissue's isolated cells (eg NSC) *in vitro*, a search for additional markers of oxidative stress (eg 4-HNE), and evidence of DNA damage, including comet assays and gammaH2AX foci, etc. Although we are not including those preliminary PAR results in this resubmission, as noted above (in the response to Rev 1) a number of additional studies have been carried out in the interim since the last submission, and a caveat can now be found in the discussion that addresses the concern about the potential mutagenic/cytotoxic effects of 4-OHT and whether these might be augmented by PNKP loss. We greatly appreciate the time and consideration that both reviewers have given to our manuscript.

Reviewer #3 (Comments to the Authors (Required)):

I dont have any further issues or concerns.

June 15, 2021

RE: Life Science Alliance Manuscript #LSA-2020-00790RR-A

Author information redacted

Dear Dr. Biernaskie,

Thank you for submitting your revised manuscript entitled "PNKP is required for maintaining the integrity of progenitor cell populations in adult mice". We would be happy to publish your paper in Life Science Alliance pending final revisions necessary to meet our formatting guidelines.

- please upload your main manuscript text as an editable doc file
- please upload your Table in editable .doc or excel format
- please be sure to include all Authors in the Author Contribution section of your manuscript
- please add your main, supplementary figure, and table legends to the main manuscript text after the references section;

Figure checks:

- please add scale bar size in legend for Figure 1I
- images in Figure S1 need scale bars, please indicate size in Legend
- please add molecular weights for blots in Figures 1G and H; S3A
- Figure S3 is not of high enough quality for publication. It must be at least 200 pixels per inch in layout

A. FINAL FILES:

- An editable version of the final text (.DOC or .DOCX) is needed for copyediting (no PDFs).
- High-resolution figure, supplementary figure and video files uploaded as individual files: See our detailed guidelines for preparing your production-ready images, <https://www.life-science-alliance.org/authors>
- Summary blurb (enter in submission system): A short text summarizing in a single sentence the

study (max. 200 characters including spaces). This text is used in conjunction with the titles of papers, hence should be informative and complementary to the title. It should describe the context and significance of the findings for a general readership; it should be written in the present tense and refer to the work in the third person. Author names should not be mentioned.

B. MANUSCRIPT ORGANIZATION AND FORMATTING:

Sincerely,

June 23, 2021

RE: Life Science Alliance Manuscript #LSA-2020-00790RRR

Author information redacted

Dear Dr. Biernaskie,

Thank you for submitting your Research Article entitled "PNKP is required for maintaining the integrity of progenitor cell populations in adult mice". It is a pleasure to let you know that your manuscript is now accepted for publication in Life Science Alliance. Congratulations on this interesting work.

DISTRIBUTION OF MATERIALS:

Again, congratulations on a very nice paper. I hope you found the review process to be constructive and are pleased with how the manuscript was handled editorially. We look forward to future exciting submissions from your lab.

Sincerely,

Eric Sawey, PhD
Executive Editor
Life Science Alliance